# An atlas of cell types in the mouse epididymis and vas deferens

**Vera D Rinaldi[1†], Elisa Donnard[2†], Kyle Gellatly[2], Morten Rasmussen[1‡], Alper Kucukural[2], Onur Yukselen[2], Manuel Garber[2,3], Upasna Sharma[4], Oliver J Rando[1]\***

[1]Department of Biochemistry and Molecular Pharmacology, University of Massachusetts Medical School, Worcester, United States; [2]Department of Bioinformatics and Integrative Biology, University of Massachusetts Medical School, Worcester, United States; [3]Program in Molecular Medicine, University of Massachusetts Medical School, Worcester, United States; [4]Department of Molecular, Cell and Developmental Biology, University of California Santa Cruz, Santa Cruz, United States

**\*For correspondence:**
Oliver.Rando@umassmed.edu

[†]These authors contributed equally to this work

**Present address:** [‡]Department of Virus and Microbiological Special Diagnostics, Statens Serum Institut, Copenhagen, Denmark

**Competing interests:** The authors declare that no competing interests exist.

**Abstract** Following testicular spermatogenesis, mammalian sperm continue to mature in a long epithelial tube known as the epididymis, which plays key roles in remodeling sperm protein, lipid, and RNA composition. To understand the roles for the epididymis in reproductive biology, we generated a single-cell atlas of the murine epididymis and vas deferens. We recovered key epithelial cell types including principal cells, clear cells, and basal cells, along with associated support cells that include fibroblasts, smooth muscle, macrophages and other immune cells. Moreover, our data illuminate extensive regional specialization of principal cell populations across the length of the epididymis. In addition to region-specific specialization of principal cells, we find evidence for functionally specialized subpopulations of stromal cells, and, most notably, two distinct populations of clear cells. Our dataset extends on existing knowledge of epididymal biology, and provides a wealth of information on potential regulatory and signaling factors that bear future investigation.

## Introduction

In mammals, the production of sperm that are competent to fertilize the egg requires multiple organs that together constitute the male reproductive tract. A common characteristic of these tissues is the presence of specialized somatic cells that support the development and maturation of gametes. For instance, spermatogenesis requires multiple functionally distinct cells in the testis that support the development and function of sperm, most notably including Sertoli cells, which envelop and provide support to developing sperm, and Leydig cells, which are located in the testicular interstitial space and are responsible for testosterone biosynthesis.

Following completion of testicular spermatogenesis, sperm are morphologically mature, but do not yet exhibit forward motility and are incapable of fertilization. After sperm exit the testis, they enter a long convoluted tube known as the epididymis, where they continue to mature over the course of approximately 10 days in the mouse. During this journey, sperm are concentrated as they proceed from caput (proximal) to corpus and then to the cauda (distal) epididymis, where they can be stored for prolonged periods in an inactive state thanks to the ionic microenvironment of the lumen. Epididymal transit has long been understood to be essential for male fertility (*Bedford, 1967*; *Hinton et al., 1996*; *Orgebin-Crist, 1967*; *Young, 1931*), and sperm undergo extensive molecular and physiological changes during this process (*Cooper, 2015*; *Cornwall, 2009*; *Gervasi and Visconti, 2017*). Most notably, sperm proteins – including, but not limited to, protamines – become

extensively disulfide-crosslinked during epididymal maturation, and this is thought to mechanically stabilize various sperm structures (*Balhorn, 1982*; *Calvin and Bedford, 1971*). At the sperm surface, lipid remodeling events include cholesterol removal and an increase in polyunsaturated fatty acids which results in increased membrane fluidity (*Saez et al., 2011*), while maturation of the glycocalyx involves extensive remodeling of accessible sugar moieties (*Tecle and Gagneux, 2015*; *Tulsiani, 2006*). In addition, a heterogeneous population of extracellular vesicles collectively known as epididymosomes deliver proteins and small RNAs – some of which may play functional roles in fertilization and early development – to maturing sperm (*Caballero et al., 2013*; *Conine et al., 2018*; *Krapf et al., 2012*; *Martin-DeLeon, 2015*; *Reilly et al., 2016*; *Sharma et al., 2016*; *Sharma et al., 2018*; *Sullivan et al., 2007*; *Sullivan and Saez, 2013*).

The organ responsible for this ever-expanding diversity of functions is a long tube – on the order of one to ten meters long when extended, depending on the species – of pseudostratified epithelium. The functions of the epididymis differ extensively along its length (*Domeniconi et al., 2016*; *Turner et al., 2003*), as for example genes such as *Rnase10* and *Lcn8* are expressed almost exclusively in the proximal epididymis, most notably in the testis-adjacent section known as the 'initial segment' (*Hsia and Cornwall, 2004*; *Jelinsky et al., 2007*; *Jervis and Robaire, 2001*; *Johnston et al., 2005*; *Turner et al., 2003*). Connective tissue septa separate the mouse epididymis into 10 anatomically defined segments (this number varies in other mammals) (*Turner et al., 2003*), with early microarray studies revealing a variety of gene expression profiles across these segments, and defining roughly six distinct gene expression domains across the epididymis (*Johnston et al., 2005*). Dramatic gradients of gene expression along the epididymis are also observed in many other mammals, including rat (*Jelinsky et al., 2007*; *Jervis and Robaire, 2001*), boar (*Guyonnet et al., 2009*), and human (*Browne et al., 2019*; *Dubé et al., 2007*; *Legare and Sullivan, 2019*; *Thimon et al., 2007*; *Zhang et al., 2006*). In addition to the variation in gene regulation along the epididymis, the epididymal epithelium at any point along the path is comprised of multiple morphologically- and functionally distinct cell types, including principal cells, basal cells, and clear cells (*Breton et al., 2016*).

Here, we sought to further explore the cellular makeup and gene expression patterns across this relatively understudied organ. In recent years, microfluidic-based cell isolation and molecular barcoding strategies, coupled with continually improving genome-wide deep sequencing methods, have enabled ultra-high-throughput analysis of gene expression in thousands of individual cells from a wide range of tissues (*Cao et al., 2017*; *Farrell et al., 2018*; *Han et al., 2018*; *Ramsköld et al., 2012*). Using droplet-based (10X Chromium) single-cell RNA-Seq, we profiled gene expression in 8880 individual cells from across the epididymis and the vas deferens. Our data confirm and extend prior studies of segment-restricted gene expression programs, and reveal that these regional programs are largely driven by principal cells. We find evidence suggesting the possibility of several novel cell subpopulations, most notably including an Amylin-positive clear cell subtype, and we predict intercellular signaling networks between different cell types. Together, these data provide an atlas of cell composition across the epididymis and vas deferens, and provide a wealth of molecular hypotheses for future efforts in reproductive physiology.

## Results

### Dataset generation and overview

We set out to characterize the regulatory program across the post-testicular male reproductive tract, focusing on four coarsely defined anatomical regions – the caput, corpus, and cauda epididymis, as well as the vas deferens (*Figure 1A*, *Figure 1—figure supplement 1*). We first generated a baseline dataset via traditional RNA-Seq profiling for at least 10 individual samples of each dissection (*Supplementary file 1*). Our bulk RNA-Seq dataset confirms the expected region-specific gene expression patterns previously observed in many species (*Figure 1B*, *Supplementary file 1*), and recapitulates, at coarser anatomical resolution but improved genomic resolution, prior microarray analyses of the murine epididymis (*Johnston et al., 2005*; *Figure 1—figure supplement 2*). Furthermore, we include below a more detailed analysis of gene expression in the vas deferens, a tissue which is not often included in published transcriptome studies.

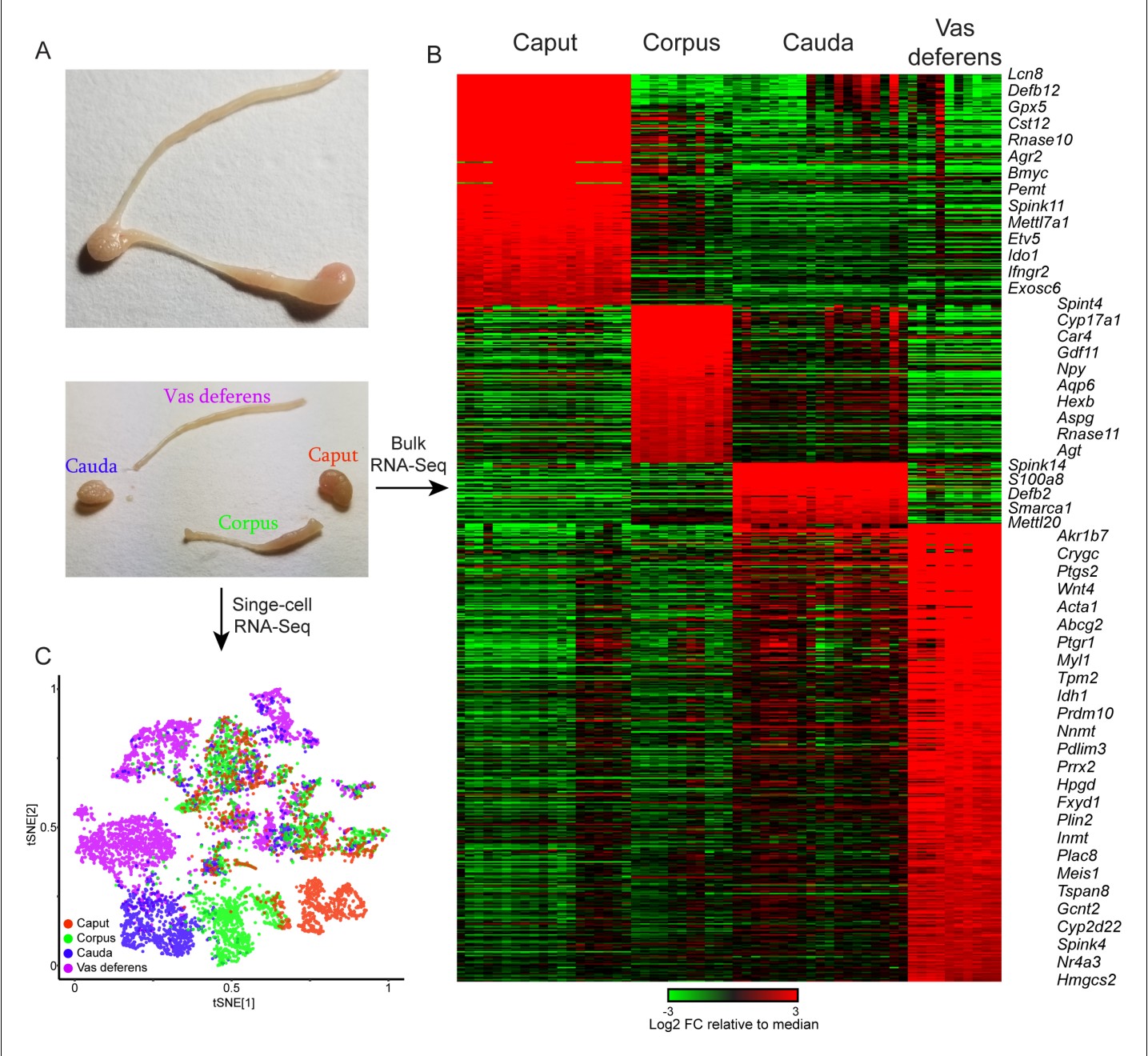

**Figure 1.** Overview of dataset.  (A) Tissue samples surveyed in this study. Top image shows a single mouse epididymis prior to dissection, while bottom shows a typical dissection into the four anatomical regions – the caput, corpus, and cauda epididymis, as well as the vas deferens –characterized by bulk RNA-Seq and single-cell RNA-Seq. See also *Figure 1—figure supplement 1*. (B) Bulk RNA-Seq dataset, showing region-enriched genes (log2 fold change relative to dataset median of at least 2, with a maximum mRNA abundance in one of the four dissections of at least 20 ppm). See also *Figure 1—figure supplement 2*. (C) Single-cell RNA-Seq dataset, clustered by t-SNE and annotated according to the four anatomical regions in (A). The online version of this article includes the following figure supplement(s) for figure 1:

**Figure supplement 1.** Light sheet imaging of mouse epididymis.

**Figure supplement 2.** Comparison of RNA-Seq to prior microarray analysis of segmental gene expression.

To disentangle the contributions of the diverse cell types comprising any given region of the mouse epididymis, we used microfluidic single-cell barcoding (10X Chromium) and single-cell RNA-Seq (scRNA-Seq) to characterize gene expression in dissociated cells from the four anatomical regions – caput, corpus, cauda, and vas deferens (*Figure 1A*, *Figure 1—figure supplement 1*). Each population represented a pool of eight tissue samples obtained from four male mice sacrificed at 10–12 weeks of age. We note that dissection margins were varied slightly between individual samples to ensure complete recovery of any cell types present at the dissection borders.

Following scRNA-Seq, quality control, and removal of reads attributable to contaminating cell-free RNA (Materials and methods), we obtained a final dataset with 8880 individual cells with an average of 3110 (median = 1965) unique molecular identifiers (UMIs) per cell. We detected cells with similar transcriptional profiles through dimensionality reduction and clustering (Materials and methods), and the resulting map was visualized using t-distributed stochastic neighbor embedding (t-SNE), with cells from each sample labeled in different colors in *Figure 1C*. Consistent with the dramatic segment-specific gene expression programs known to distinguish the different regions of the epididymis, we find that roughly half of the clusters are composed entirely of cells from one tissue sample – the caput, corpus, cauda, or vas (*Figure 1C*). Region-specific clusters included all principal cell clusters (discussed below), as well as muscle cells and a subset of stromal cells, both of which were enriched in the vas deferens samples. The remaining cell clusters included cells from multiple dissections and thus represent cell types present throughout the epididymis.

We assigned the 21 cell clusters to known cell types by the expression profiles of marker genes detected through differential expression analysis (*Figure 2A*, *Figure 2—figure supplement 1*, *Supplementary file 2*). This identified several populations of principal cells and basal cells, one of clear cells, and multiple groups of support cells including fibroblasts, endothelium, muscle, and immune cells (*Figure 2A*). Characteristic genes enriched in each cluster are shown in *Figure 2B*, where we highlight both known and novel marker genes for various epididymal cell types. To expand on gene enrichment in cell clusters and to visualize gene expression across all cells in our dataset, *Figure 2C* shows expression levels of several well-known marker genes, as well as a handful of genes of interest for downstream analyses, across all individual cells. Overall, we find that epididymal cells are distinguished primarily by two broad molecular features. First, functionally and histologically distinct cell types such as clear cells, basal cells, and so on are characterized by known marker genes, as for example genes encoding vacuolar ATPase subunits (*Atp6v1a*, *Atp6v0c*, etc.) are highly expressed in clear cells which are responsible for luminal acidification (*Breton et al., 1998*; *Breton et al., 1999*; *Brown et al., 1992*). Second, even for a relatively histologically uniform cell type, the principal cells, a large number of genes distinguish cells originating in different regions along the epididymal tube. This suggests that principal cells are specialized within their niches, maintaining a complex succession of luminal microenvironments across the epididymis.

Below, we provide an overview of the various cell populations across the epididymis, detailing the molecular features of each cell type identified, and further exploring cellular diversity.

## Segment-specific gene expression in principal cells

Principal cells are highly active secretory and absorptive cells responsible for producing much of the unusual protein composition of the epididymal lumen, as well as playing additional roles in modulating luminal pH and in lumicrine signaling. As their name suggests, principal cells comprise the major cell type present in the adult epididymis, and our initial principal cell clusters could be readily assigned to anatomically defined segments by comparison to the gold standard microarray analysis of epididymal gene expression (*Johnston et al., 2005*). To this initial set of six epididymal clusters, we added three clusters of epithelial cells from the vas deferens (*Figure 3A*), which appear in our dataset (based on shared markers) to be analogous to epididymal principal cells.

To investigate cellular heterogeneity among principal cells, we performed iterative re-clustering with cells from these nine clusters (Materials and methods, *Figure 3B*, *Figure 3—figure supplement 1*, *Supplementary file 3*). The second round of clustering was again dominated by the anatomical origin of principal cells, separating cells from the caput, corpus, cauda, and vas deferens (*Figure 3C*). Moreover, at this resolution, cells from each anatomic region could be further subdivided into three to four subtypes (*Figure 3B*), largely reflecting further subspecialization of cells in the various compartmentalized epididymal segments (*Johnston et al., 2005*). For instance, cells corresponding to the mouse initial segment were readily identified as a subset of principal cells from

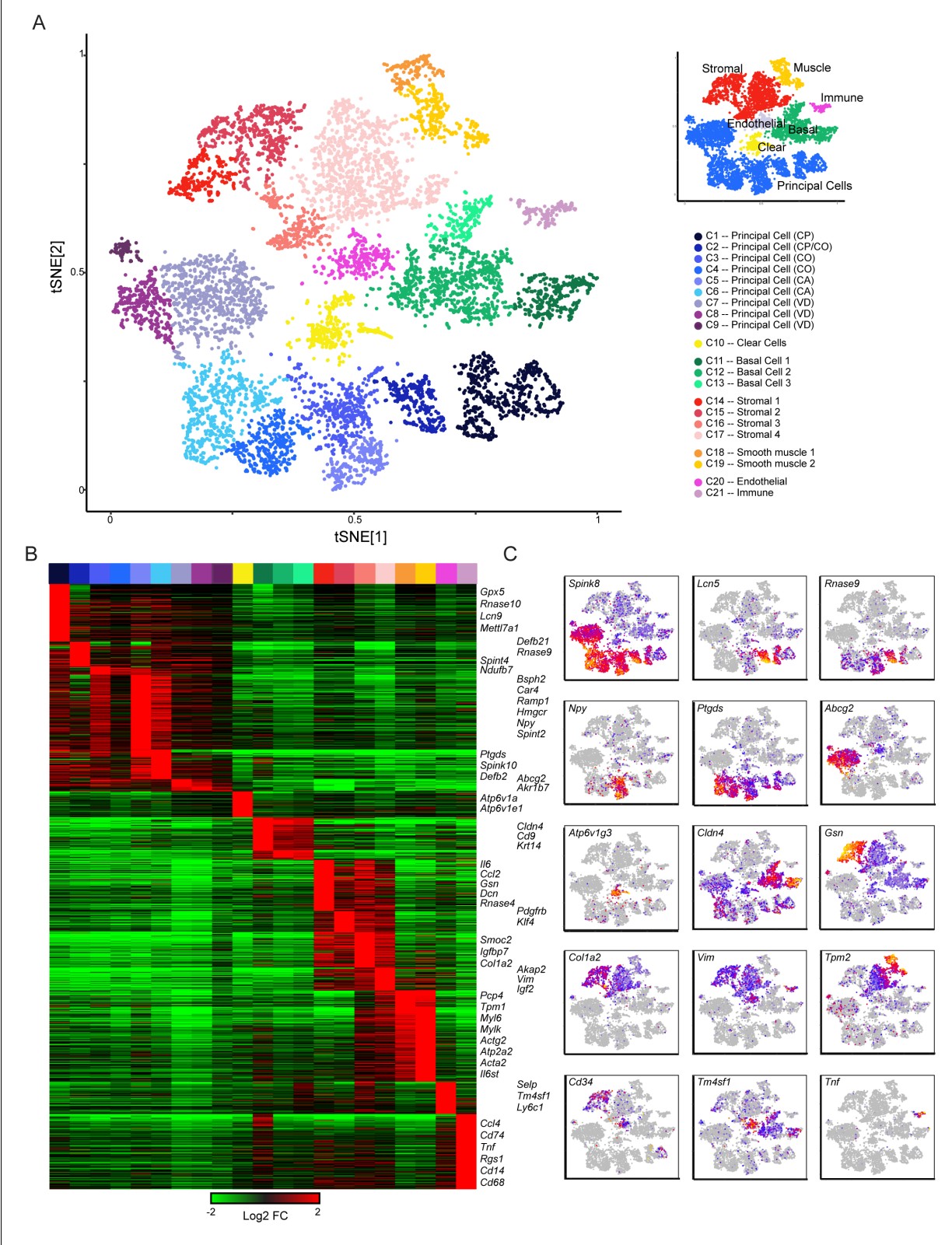

**Figure 2.** Single-cell decomposition of the epididymis. (**A**) Single-cell cluster (expanded from *Figure 1C*) with 21 clusters annotated according to predicted cell types. Inset shows coarser cell type groupings used for downstream subclustering. (**B**) Heatmap of genes enriched in each cluster. Among the genes exhibiting significant variation across the 21 clusters, 1241 genes enriched at least fourfold in at least one cluster were selected for visualization. Heatmap shows these genes sorted according to the cluster where they exhibit maximal expression. (**C**) Expression of key marker genes

*Figure 2 continued on next page*

*Figure 2 continued*

across the entire dataset. See also *Figure 2—figure supplement 1*. For this and other single-cell images, color scale runs from gray (no expression) through blue, then red and finally orange (highly expressed), with heatmap for each panel scaled according to expression level of the gene in question. The online version of this article includes the following figure supplement(s) for figure 2:

**Figure supplement 1.** Additional images of gene expression levels across the entire single-cell dataset.

the caput dissection based on their expression of genes (*RNase10*, *Cst11*, *Lcn2*, *Mfge8*, and others) previously associated with segment 1 (*Johnston et al., 2005*), and were distinguished from other cells from more distal segments of the caput epididymis (*Lcn2* and *Defb15* for segments 3–4, etc.). Similarly, corpus and cauda principal cell subclusters reflected anatomical segmentation (*Figure 3D–E*), separating corpus cells into segments 5–6 and 7 (*Lcn5/Rnase9/Plac8* vs *Npy*, respectively), and cauda cells into segments 8–9 and 10 (*Gpx3/Klk1b27/Hint1/Gstm2* vs *Ptgds/Spink10/Crisp1*). Also consistent with prior data, while some genes were focally expressed in only one or two adjacent segments (eg *Npy* for segments 7–8), other spatially restricted genes were expressed across broader domains crossing multiple contiguous segments (eg *Ptgds*, *Defb43*). Overall, our subclustering is concordant with prior functional and molecular studies of epididymal segmentation, and supports the notion that the anatomical segmentation of the epididymis generates multiple distinct microenvironments that sperm experience during post-testicular maturation (*Domeniconi et al., 2016*).

As is clear from many prior studies of epididymal gene regulation (*Guyonnet et al., 2009*; *Hales et al., 1980*; *Hall et al., 2007*; *Jelinsky et al., 2007*; *Jervis and Robaire, 2001*; *Johnston et al., 2005*; *Papp et al., 1995*; *Ribeiro et al., 2016*; *Thimon et al., 2007*), the genes that define principal cells from any given segment tend to be members of multi-gene families encoded in genomic clusters of many paralogous genes (*Figure 3D–E*, *Figure 3—figure supplement 2*). These clusters include gene families encoding serine protease inhibitors, antimicrobial defensin peptides, cysteine-rich peptides involved in glycocalyx maturation, small molecule-binding lipocalins, aldo-keto reductases, and RNaseA family members. The encoded proteins play roles throughout the male reproductive tract, both locally in the epididymal lumen as well as in downstream regions of the male reproductive tract, or even in the female reproductive tract (see Discussion). Curiously, a number of these region-specific proteins, including various metabolic enzymes, small-molecule-binding proteins, and defensins, also end up decorating the sperm surface.

As our data largely confirm the extensive prior literature on segmental gene regulation across the epididymis, we focused next on gene expression in the vas deferens, which has been the subject of relatively few genome-wide studies (*Snyder et al., 2010*). In contrast to the principal cells of the epididymis, few of the marker genes for the vas deferens are encoded in large multi-gene families, with two major exceptions being members of gene families encoding serine protease inhibitors (*Akr1b7*, *Akr1c1*) (*Jagoe et al., 2013*) and crystallin proteins (*Crygb*, *Crygc*). Instead, genes enriched in the vas deferens included various transmembrane transporters (*Abcg2*, *Fxyd4*, *Aqp2*, etc.) and signaling molecules (*Plac8*, *Ptgr1*, *Ptges*, *Ptgs2*, *Prlr*, *Cited1*, *Pcbd1*, *Comt*, *Slco2a1*, etc.). Vas deferens clusters were also enriched for expression of nuclear-encoded genes involved in mitochondrial energy production (*Atp5g1*, *Uqcr10*, *Ndufa3*, *Cox6b1*, *Ndufa7*, *Ndufa11*, *Atp5e*, *Mrpl33*, etc.), but this was not unique to the vas deferens – across the epididymis, mitochondrial energy production was enriched in principal, clear, and muscle cells, and depleted in basal, immune, endothelial, and stromal cells (*Figure 3—figure supplement 3*).

We were particularly interested in one notable marker of vas deferens epithelial cells, *Abcg2*, which encodes a xenobiotic transporter whose functions are best-understood in the placenta. Abcg2 is localized to the apical end of syncitiotrophoblast cells of the human placenta, where it protects the developing fetus by pumping small molecules back into the maternal circulation, thereby preventing xenobiotics from crossing the placental barrier (*Wang et al., 2006*). Abcg2 has also been implicated in sperm function, as it is localized on the sperm surface where it is proposed to play a role in modulating the levels/localization of specific membrane lipids (*Caballero et al., 2012*). Although the function of Abcg2 in the vas deferens is unclear, we note that ABC-class transporters such as Abcg2 are typically localized to the apical end of polarized epithelia, which we confirmed with immunofluorescence (*Figure 3—figure supplement 4*). This suggests that Abcg2 expression in the vas deferens would be predicted to pump small molecules *into* the lumen, rather than clearing

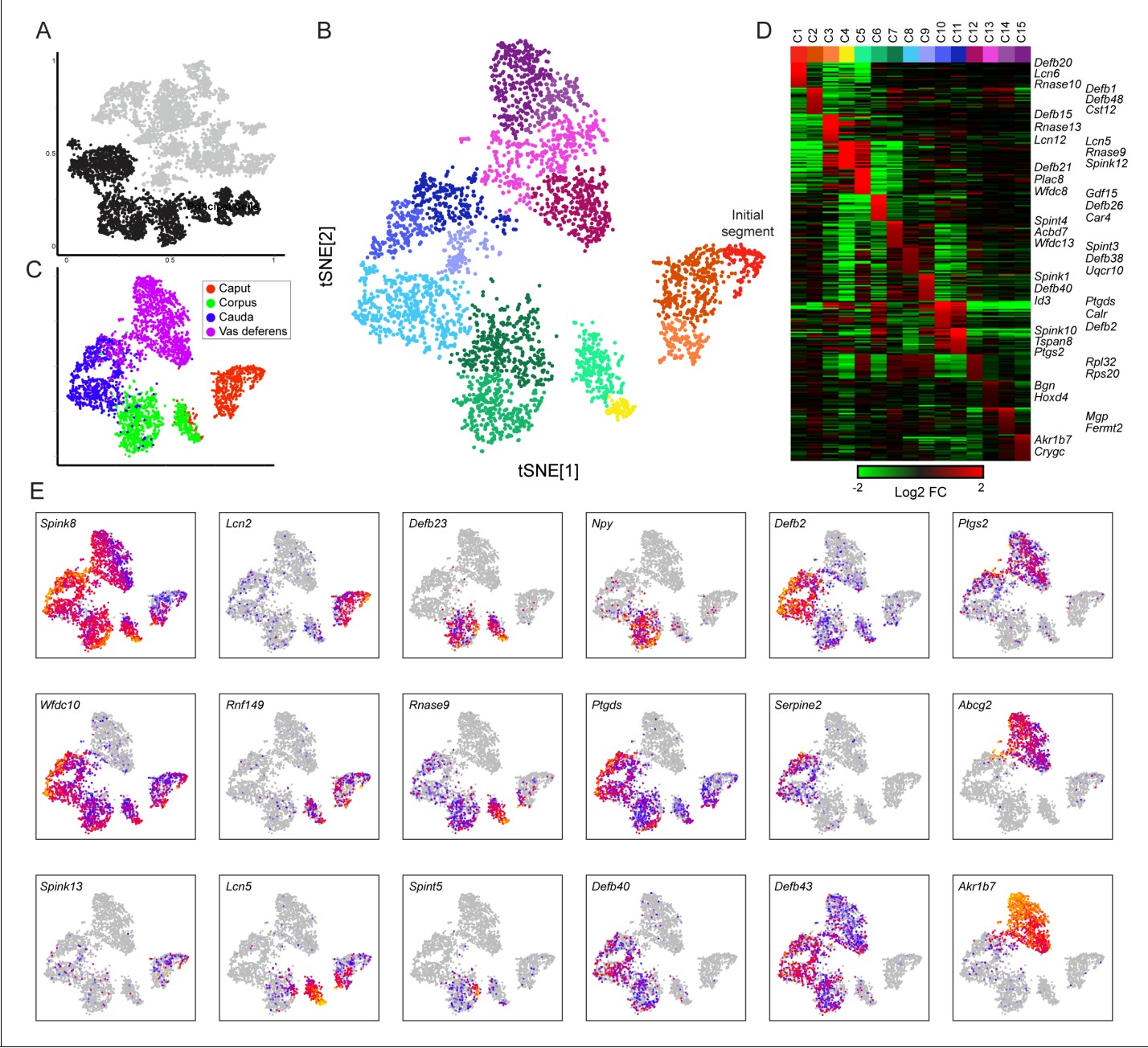

**Figure 3.** Modulation of the principal cell transcriptome across the male reproductive tract. (**A**) Clusters from the overall dataset (*Figure 2A*), with the principal cell clusters extracted for reclustering highlighted in black. (**B**) Reclustering of extracted principal cells, visualized by t-SNE. Distinct colors highlight the 15 resulting principal cell clusters. See also *Figure 3—figure supplements 1–4*. (**C**) Reclustered principal cells, as in panel (**B**), with cells colored according to anatomical origin. (**D**) Heatmap showing the top 20 genes enriched in each of the principal cell clusters. Based on highly enriched marker genes, our epididymis clusters from 1 to 11 (ignoring the four vas deferens clusters) correspond to the following segments from *Johnston et al., 2005*: segment 1 (initial segment), segments 1–2, segments 3–4, segment 5 (early), late segment 5/segment 6, segment 7, late segment 7, segments 8–9, segment 9, late segment 9 (some segment 10 markers), segment 10. (**E**) Expression of key marker genes across all principal cells, visualized as in *Figure 2C*.

The online version of this article includes the following figure supplement(s) for figure 3:

**Figure supplement 1.** Gene expression heterogeneity within principal cell subclusters.

**Figure supplement 2.** Region-specific expression of multi-gene family members.

**Figure supplement 3.** High levels of oxidative energy production in principal, clear, and muscle cells.

**Figure supplement 4.** Apical localization of Abcg2 in the vas deferens.

them out. As ABC class transporters such as Abcg2 are known to pump some endogenously synthesized molecules (such as heme), our data suggest an unappreciated role for this protein in concentrating some factor(s) in the lumen of the vas deferens.

## Clear cells

Clear cells are largely responsible for one of the best-known functions of the epididymis, which is to provide an acidic luminal environment that ensures that sperm remain quiescent until mating occurs (*Breton et al., 1998*; *Breton et al., 1999*; *Brown et al., 1992*). This function is accomplished via high-level expression of vacuolar ATPase (V-ATPase) genes (*Breton and Brown, 2013*; *Breton et al., 1996*; *Carr and Acott, 1984*; *Hermo et al., 2000*), and clear cells are readily identified in our dataset by expression of the V-ATPase-encoding marker genes *Atp6v0c*, *Atpv1e1*, and *Atp6v1a* (*Figure 2B–C*). In addition to expressing genes encoding V-ATPase subunits and V-ATPase-interacting proteins (*Dmxl1*), clear cells were enriched for genes involved in a wide range of functions, with the two most prominent being energy metabolism and membrane trafficking. In the former case, similar to vas deferens principal cells, clear cells also express high levels of genes involved in metabolic energy production, presumably serving to provide the high levels of ATP needed to power the acidification of the epididymal lumen. In addition to this elevated energy production, clear cells are also characterized by high expression of various membrane trafficking genes (*Dab2*, *Ap1s3*, *Arf3*, *Stx7*, *Cdc42se2*). This is consistent with the important role for active membrane recycling/endocytosis in localization of the vacuolar ATPase to clear cell microvilli (*Breton and Brown, 2013*; *Shum et al., 2011*).

To explore diversity among epididymal clear cells, we isolated and reclustered them (*Figure 4A*) as described above for principal cells, finding three clear cell subclusters (*Figure 4B*). Two of these clusters included clear cells from throughout the epididymis, while Cluster 2 was comprised entirely of clear cells from the caput and corpus epididymis (*Figure 4C*). This subcluster was distinguished by the expression of a small number of genes, with one standout marker gene: *Iapp*, encoding islet amyloid polypeptide, or Amylin (*Figure 4D*, *Figure 4—figure supplement 1*). To validate these potentially distinct clear cell types, we stained epididymal tissue sections for a general clear cell marker (V-ATPase) and for Amylin. Consistent with the single-cell expression profiles, we find that

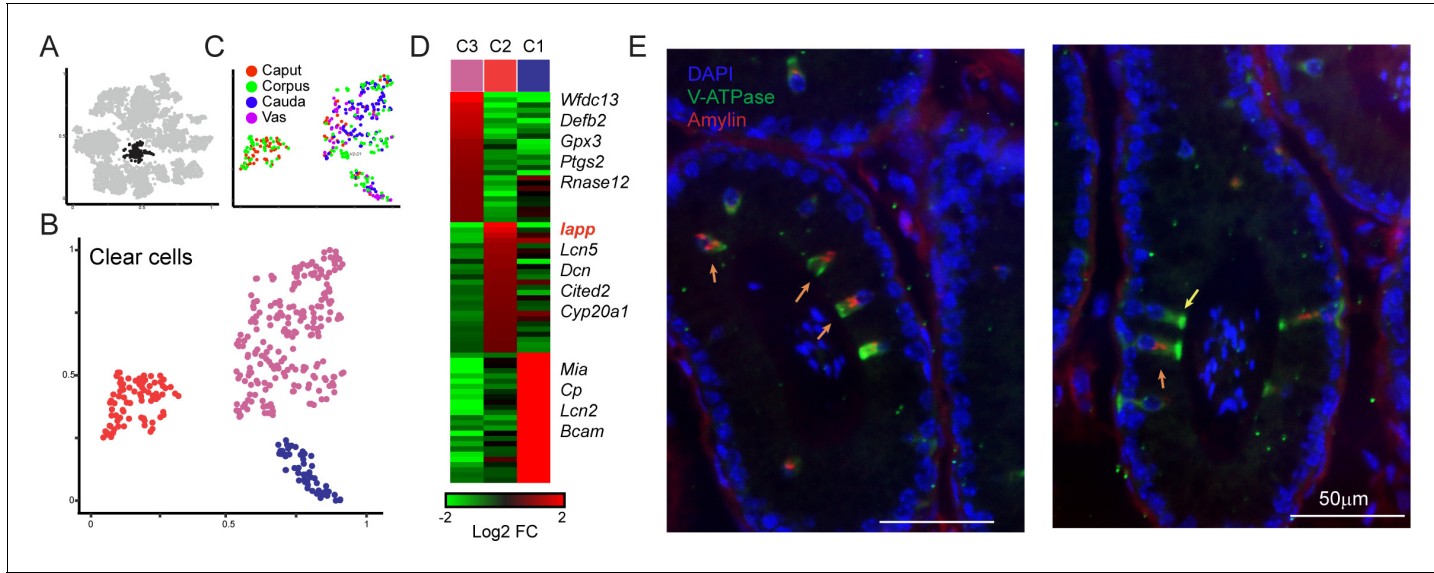

**Figure 4.** Amylin is a marker for a distinct subset of clear cells. (A-C) Reclustering of clear cells. Panels are analogous to *Figure 3A–C*. (D) Heatmap showing marker genes for the three clear cell clusters. (E)Amylin marks a subset of epididymal clear cells. Immunofluorescence image of a tubule of the caput epididymis, stained for DAPI, V-ATPase-G3 (marker for clear cells), and Amylin, as indicated. Yellow arrow shows an Amylin-negative clear cell, while orange arrows show Amylin-positive clear cells. See also *Figure 4—figure supplement 1*.

The online version of this article includes the following figure supplement(s) for figure 4:

**Figure supplement 1.** Clear cell subspecialization.

Amylin labels a subset of V-ATPase-positive clear cells in the caput and the corpus epididymis, and that these cells were undetectable in the cauda or vas deferens (*Figure 4E*, *Figure 4—figure supplement 1C* and not shown). We note that it seems unlikely that Amylin-positive clear cells correspond to the classical clear cell subtypes known as 'narrow' and 'apical' cells, which are uniquely found in the initial segment of the caput epididymis (*Abou-Haila and Fain-Maurel, 1984*; *Adamali and Hermo, 1996*; *Breton et al., 2016*; *Sun and Flickinger, 1979*; *Sun and Flickinger, 1980*), as we find Amylin-positive cells outside of the initial segment. Together with the observation that only a subset of clear cells in any given cross-section were Amylin-positive, our data suggest the intriguing possibility that Amylin-positive cells represent a functionally distinct subclass of clear cells.

## Basal cells

The third major cell type consists of the basal cells, so-named for the fact that their cell bodies are primarily localized on the basal surface of the tube, although their cytoplasmic processes have been shown to reach the lumen in some regions of the epididymis (*Shum et al., 2008*). Basal cells are believed to play an important role in maintaining the structural integrity of the blood-epididymis barrier, and it has been proposed that they may be adult stem cells for the epididymal epithelium (*Mandon et al., 2015*; *Pinel et al., 2019*). Three clusters of basal cells were identified in our dataset based on the shared expression of marker genes including *Itga6* and *Krt14* (*Figure 2C–D*). Co-expressed with these genes were a wide variety of genes involved in cell adhesion (*Bcam*, *Gja1*, *Cldn1*, *Cldn4*, *Epcam*, *Sfn*, *Cdh16*, *Itgb6*) and intercellular signaling (*Adm*, *Egfr*, *Hbegf*, *Nrg1*, *Ctgf*, *Dll1*, *Egfl6*, *Cd44*, *Cd9*, *Tacstd2*), consistent with a recent genome-wide analysis of isolated rat basal cells (*Mandon et al., 2015*). In addition to expressing cell adhesion molecules, basal cells exhibited high level expression of several genes involved in membrane trafficking and lipid metabolism (*Cd9*, *Sdc1*, *Sdc4*, *Vmp1*, *Apoe*, *Apoc1*, *Sgms2*), suggesting that basal cells may exhibit more dynamic membrane remodeling, or vesicle production, than currently appreciated.

It has been suggested that basal cells of the epididymis express antigens common to mononuclear phagocytic cells such as macrophages and dendritic cells and thus may derive from these cells or even overlap with them functionally (*Seiler et al., 1999*; *Seiler et al., 1998*; *Yeung et al., 1994*). To explore the relationship between these cell types in our dataset, we extracted all putative basal and immune cells for reclustering. We observe a clear separation between basal cells and immune cells based on gene expression patterns (*Figure 5—figure supplement 1A–B* and see below). Furthermore, our followup co-staining studies with basal cell markers and mononuclear phagocyte markers were consistent with prior reports (*Shum et al., 2014*) showing that basal cells are separate from macrophages and other antigen-presenting cells (*Figure 5—figure supplement 1C–D*). Together, our scRNA-Seq data and histology studies confirm that differentiated basal cells are distinct from other cells of the monocyte/macrophage lineage.

To further explore the diversity of basal cell subtypes in our dataset, we extracted and reclustered the three putative basal cell clusters as described above for principal and clear cells. This analysis revealed little diversity among basal cells (*Figure 5A–C*). Although we did observe one major subgrouping of basal cell subclusters based on expression of ribosomal protein genes (RPGs), the two basal cell subpopulations (RPG high and RPG low) also exhibited significant differences in the number of UMIs captured per cell (*Figure 5C–E*). Therefore, it is unclear whether the RPG-low basal cells represent a biologically meaningful population marked by low biosynthetic activity, or whether these cells represent technical artifacts (of, say, inefficient lysis and RNA recovery). Beyond this major grouping of basal cells, we did identify one additional cluster of possible interest, with a subset of basal cells expressing high levels of *Notch2*, *Spry2*, *Pecam1*, *Slc39a1*, *Ccl7*, *Cd93*, and *Penk*, although our followup staining efforts to identify this putative basal cell subset were inconclusive (not shown). Nonetheless, this basal cell subgroup may potentially represent an interesting topic for future followup studies.

## Supporting cells: stromal, endothelial, immune, and muscle cells

In addition to the major cell types that comprise the lining of the tube and shape sperm maturation, the epididymis is surrounded by an array of support cells. Eight of the clusters in our initial analysis clearly represent these various supporting cell types. Two clusters of muscle cells, defined by their expression of various actomyosin cytoskeletal genes (*Myh11*, *Mylk*, *Acta1*, *Tpm2*, etc.) are identified,

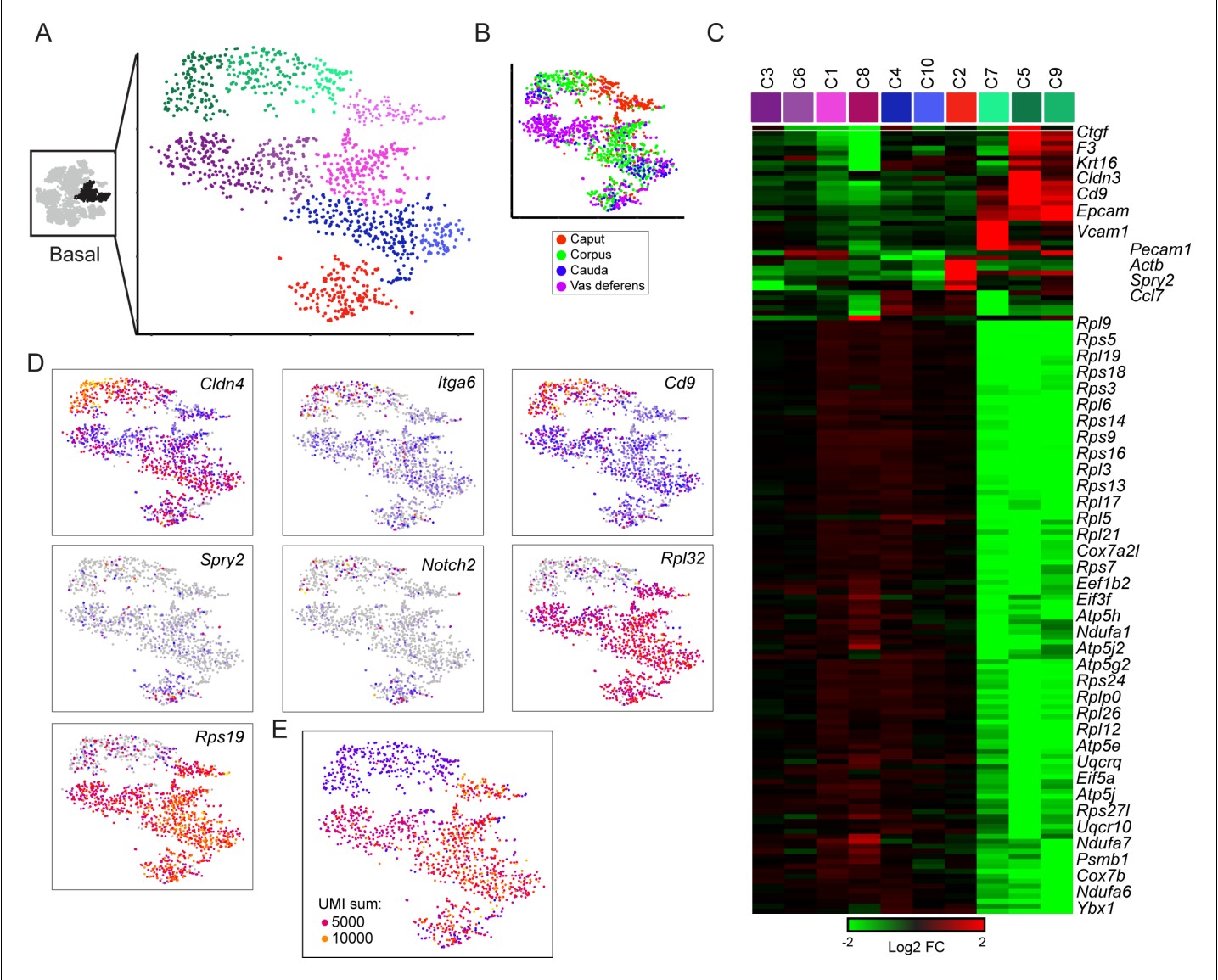

**Figure 5.** Subclustering of basal cells. (A-B) Subclustering of basal cells; (A) shows 10 clusters annotated by color, while (B) shows basal cells colored by anatomic origin. (C) Expression of marker genes (minimum fold-enrichment > 4-fold in at least one cluster) across the 10 basal cell subclusters, grouped by hierarchical clustering. Notable here are two major populations, distinguished by expression of ribosomal protein genes. A third minor cluster (C2) was marked by elevated expression of several genes including Notch2 and Spry2. (D) Expression of individual genes across the basal cell subclusters, as in *Figure 2C*. (E) UMI counts for the basal cell subclustering, revealing a strong correspondence between the RPG high/low divisions and cells with high/low UMI counts.

The online version of this article includes the following figure supplement(s) for figure 5:

**Figure supplement 1.** Clear distinctions between basal cells and various immune cell populations.

and are primarily populated by cells from the vas deferens (*Figures 1C* and *2*) where contraction of the muscular sheath drives sperm movement during ejaculation. Muscle markers are also expressed at lower levels in a subset of the collagen-enriched stromal cell clusters (*Figure 2C*), presumably highlighting a population of myofibroblasts (see below). Other supporting cell types included fibroblasts, endothelial cells from blood and lymph vessels, macrophages, and other immune cells.

To further explore the cellular diversity among the support cells, we isolated cells from the sets of clusters corresponding to muscle, stroma, immune, and endothelial cells, and subjected each group of cells to a second iteration of clustering. We found relatively little variation among the muscle cells, which separated into two clusters (*Figure 6—figure supplement 1*). These clusters were

distinguished by different expression levels of ribosomal protein genes, similar to the major distinction between basal cells with high and low UMI coverage (*Figure 5E*). Although we did not observe substantially different UMI coverage between muscle cell subclusters (not shown), given the potential for this subclustering to be a cryptic artifact rather than biological signal we did not further evaluate muscle cells.

We turned next to immune and endothelial cells, which we grouped together for purposes of visualization, as the resulting subclusters were more coherent than those obtained by subclustering either group in isolation. Here, subclustering revealed a range of infiltrating immune cells (*Figure 6A–C*), and two groups of endothelial cells, as follows:

- Macrophages expressing *Ccl3, Cll4, Cd14, Il1rn, Ifnb1, Tnf*, etc.
- A distinct group of potentially anti-inflammatory Macrophages, expressing high levels of *Trem2* and *Wfdc17*, as well as *Ctsd, Dcxr, Ctsa, Ctsb, Csf1r, Rnase4,* and *Lyz2*.
- Dendritic cells expressing *Ccr7, Kit, Gm2a, Btla*, and various MHC genes (*H2-0a, H2-Aa, H2-Ab1, H2-DMa*).
- T cells, expressing *Cd3e, Cd3g, Cd3d, Il2rb, Lck, Cd7*, and *Gzmb*.
- Endothelial cells, expressing *Cd34, Tm4sf1, Icam2*, and *Emcn*. Endothelial cells further separated into three subclusters distinguished by different intercellular signaling and adhesion molecules (*Kalucka et al., 2020*). While cluster C1 has a clear signature of capillary blood vessels and angiogenesis (*Cxcl12, Egf17, Dll4, Ly6a, Ly6c1, Rgcc, Plk2, Esm1*) and includes arterial gene markers (*Igfbp3, Sox17, Edn1*), endothelial cells in clusters C4 and C7 express genes that are markers of larger blood vessels (*Vwf, Vcam1*) as well as venous signature genes (*Selp, Slc38a5, Penk*). Additionally, clusters C4 and C7 include endothelial cells with a lymphatic gene expression signature (Cp, Fxyd6, Thy1, Prox1), although venous and lymphatic cells did not subcluster into distinct groups.

The presence of both blood and lymph vessels was expected from prior studies (*Guiton et al., 2019*; endothelial cells are not further considered here. As for immune cells, the immune populations identified by scRNA-Seq are generally concordant with previous histological and FACS-based analyses which show that macrophages, dendritic cells, and infiltrating lymphocytes represent the predominant groups of immune cells involved in the blood-epididymis barrier (*Da Silva and Barton, 2016*; *Nashan et al., 1989*; *Serre and Robaire, 1999*; *Sun and Flickinger, 1979*; *Voisin et al., 2018*). Beyond recovering these coarse immune cell subsets, we did not have sufficient cell numbers in this dataset to find the small number of anticipated B cells, or to distinguish the subpopulations of T cells present.

We finally turned to the remaining four clusters from the full dataset that we annotated as stromal cells based on their expression of various collagen genes (*Figure 2B*). After reclustering (*Figure 6D–F*), we find that the largest subcluster of these cells was characterized by high levels of muscle markers (albeit not as high as bona fide muscle cells – *Figure 2C*) such as *Tagln, Myl9, Tpm2, Acta2*, etc., and potentially corresponds to the myoid cells whose contraction plays roles in moving sperm and fluid through the epididymal lumen (*Oliveira et al., 2016*). Two other clusters of fibroblasts (C2, C4) were distinguished by relatively few marker genes and were not further considered. Finally, we identified two particularly interesting stromal cell subtypes that were almost entirely confined to the vas deferens (*Figure 6D*, inset). One of these clusters expressed a number of markers associated with stem cells (*Angpt1, Lgr5, Tspan8*) and may represent mesenchymal stem cells. The other vas-specific cluster was distinguished by high level expression of a large number of intercellular signaling molecules (*Il6, Cxcl1, Cxcl2, Cxcl12, Ccl2, Ccl7, Vcam1, Igf1, Ptx3, Tslp*, etc.) and so were tentatively annotated as 'secretory fibroblasts.' Followup staining studies confirm the presence of these two distinct cell populations in the vas deferens stroma, with a layer of Lgr5-positive stromal cells at the periphery of the vas, contrasting with extensive Ptx3/Tslp staining observed closer to the lumen (*Figure 6—figure supplement 2*). It will be interesting in future studies to investigate the biology and functions of these stromal cell populations in the vas deferens.

## Cell-to-cell communication pathways in the epididymis

Finally, we noted that many of the cell subpopulations defined throughout this study were distinguished by expression of intercellular signaling molecules, from adhesion molecules to secreted cytokines and their receptors. Prior studies decomposing complex tissues into scRNA-Seq have leveraged the expression profiles for known ligand-receptor pairs to infer significant pathways for cell-

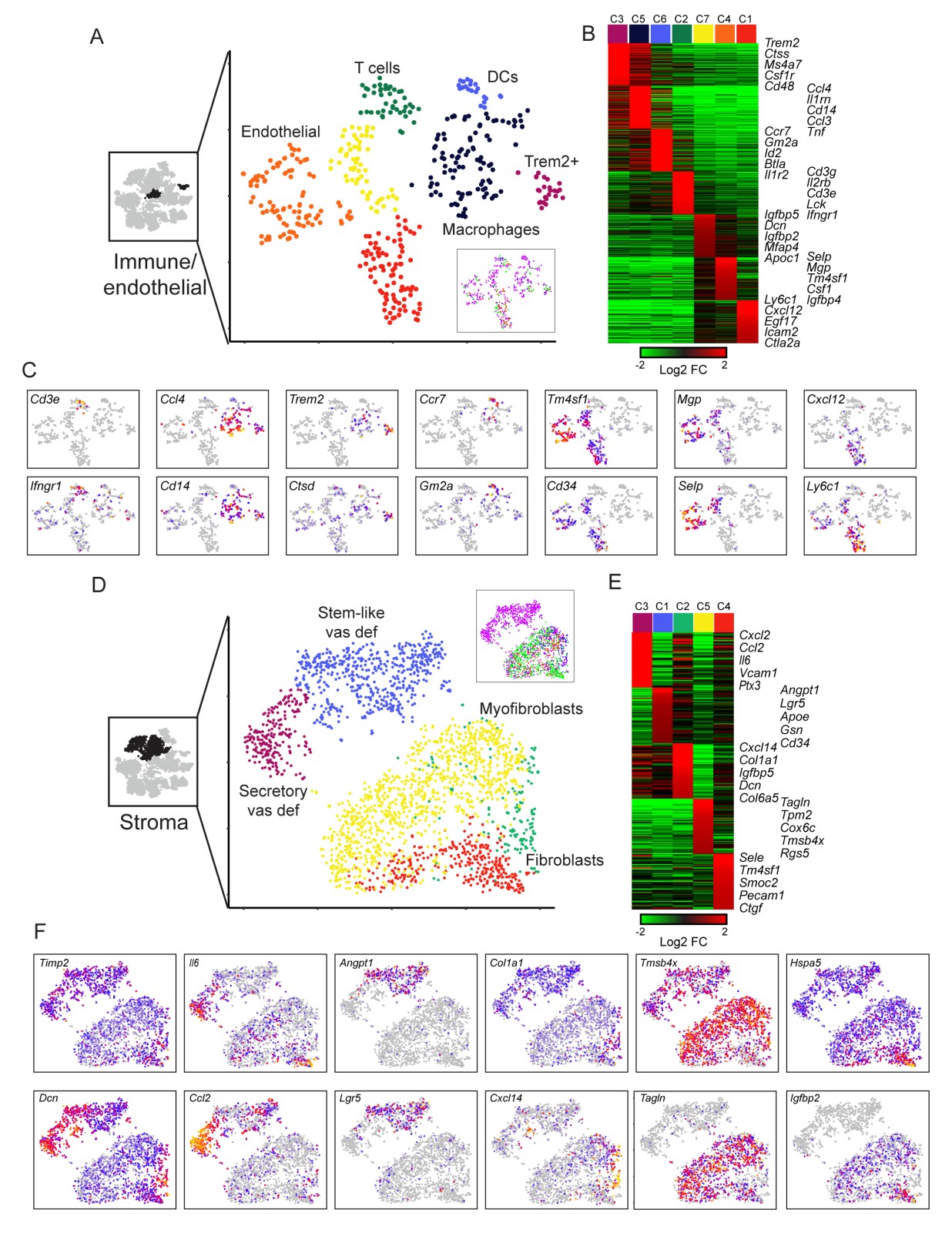

**Figure 6.** Diverse stromal cell populations in the epididymis. (**A**) Reclustering of immune and endothelial cells. Left panel highlights the clusters from full dataset used for reclustering, while right panels show reclustered data. Inset shows cells colored according to the anatomical region where they were found. (**B**) Heatmap of the top 100 markers for each of the immune and endothelial subclusters in (A). (**C**) Expression of individual marker genes

*Figure 6 continued on next page*

*Figure 6 continued*

across immune and endothelial cell subpopulations, as in *Figure 2C*. (D-F) Reclustering of stromal cells, arranged as in panels (A-C). Heatmap (E) shows the top 50 genes for each subcluster. See also *Figure 6—figure supplement 2*.

The online version of this article includes the following figure supplement(s) for figure 6:

**Figure supplement 1.** Subclustering of muscle cells.

**Figure supplement 2.** Validation of secretory and stem-like stromal cells in the vas deferens.

to-cell communication (*Cohen et al., 2018*). Motivated by these prior efforts, we asked whether our dataset could be used to explore potential intercellular communication networks throughout the epididymis.

We identified known ligands, and their receptors, expressed across the 34 cell subpopulations defined in this study (*Supplementary file 4*), using a large-scale ligand-receptor resource (*Ramilowski et al., 2015*). From these lists, we identified the number of potential interactions linking every cell type to every other cell type (*Figure 7*). Of these, significant interactions (p<0.01) between cell types were defined based on the number of cases of expression of a ligand-receptor pair between the cell types, relative to the number of pairs expected by chance based on the total number of ligands and receptors expressed in each cell type (Materials and methods). This revealed multiple potential pairs of cells in communication, of which we highlight two. First, we find that basal cells and endothelial cells expressed complementary sets of ligands and receptors, consistent with the spatial juxtaposition between basal cells and the subepithelial capillary beds (*Abe et al., 1984*). Second, we found reciprocal connections between macrophages and the 'secretory fibroblasts' in the vas deferens, suggesting the possibility that these vas deferens stromal cells play a role in recruiting macrophages. Indeed, although we found macrophages throughout the epididymis and vas deferens (*Figure 6A*), there was a bias for macrophages to be found in the vas deferens sample in our dataset (p<$10^{-7}$, hypergeometric). Our data thus highlight candidate factors to be investigated in future genetic studies that aim to explore the role of signaling pathways in cell-to-cell communication within the epididymis.

## Discussion

Here, we present a single-cell atlas of the murine epididymis and vas deferens, building on prior histological and marker-based surveys of the epididymis. We recover all previously described cell types in the murine epididymis, including support cells which had not been characterized in detail in this tissue. We document more complete molecular profiles for the individual cell populations and provide a compendium of transcriptional profiles for a range of cells from this key reproductive organ. We further identify markers for potentially novel cell types, and our analyses suggest a wide variety of new hypotheses for future molecular studies.

### Distinctive principal cell programs across the epididymis

The most striking feature of the epididymis is the presence of regional gene expression programs in the principal cells that comprise the majority of epididymal epithelial cells. Consistent with previous surveys of the epididymis in a variety of mammals, we find multiple discrete groups of principal cells arrayed along the epididymis, with roughly eight unique gene expression signatures being appreciable across the epididymis and vas deferens (*Figure 3D*). Even within a given cluster, we find evidence for heterogeneous expression of marker genes between individual cells (*Figure 3—figure supplement 1*, *Supplementary file 3*), consistent with immunostaining studies documenting patchy expression of GSTs and other proteins among principal cells in a given epididymal region (*Hermo et al., 1992*; *Hermo et al., 1991*; *Papp et al., 1995*; *Rankin et al., 1992*). Importantly, these highly variable markers exhibit little co-expression – this cellular heterogeneity is insufficient to drive further subclustering – suggesting that these cell-to-cell differences arise from stochastic expression of individual genes rather than coordinated expression of a broader program. Whether these heterogeneous expression patterns reflect temporally stable subspecialization/individualization of principal cells, or whether individual cells in a given region all express the same characteristic

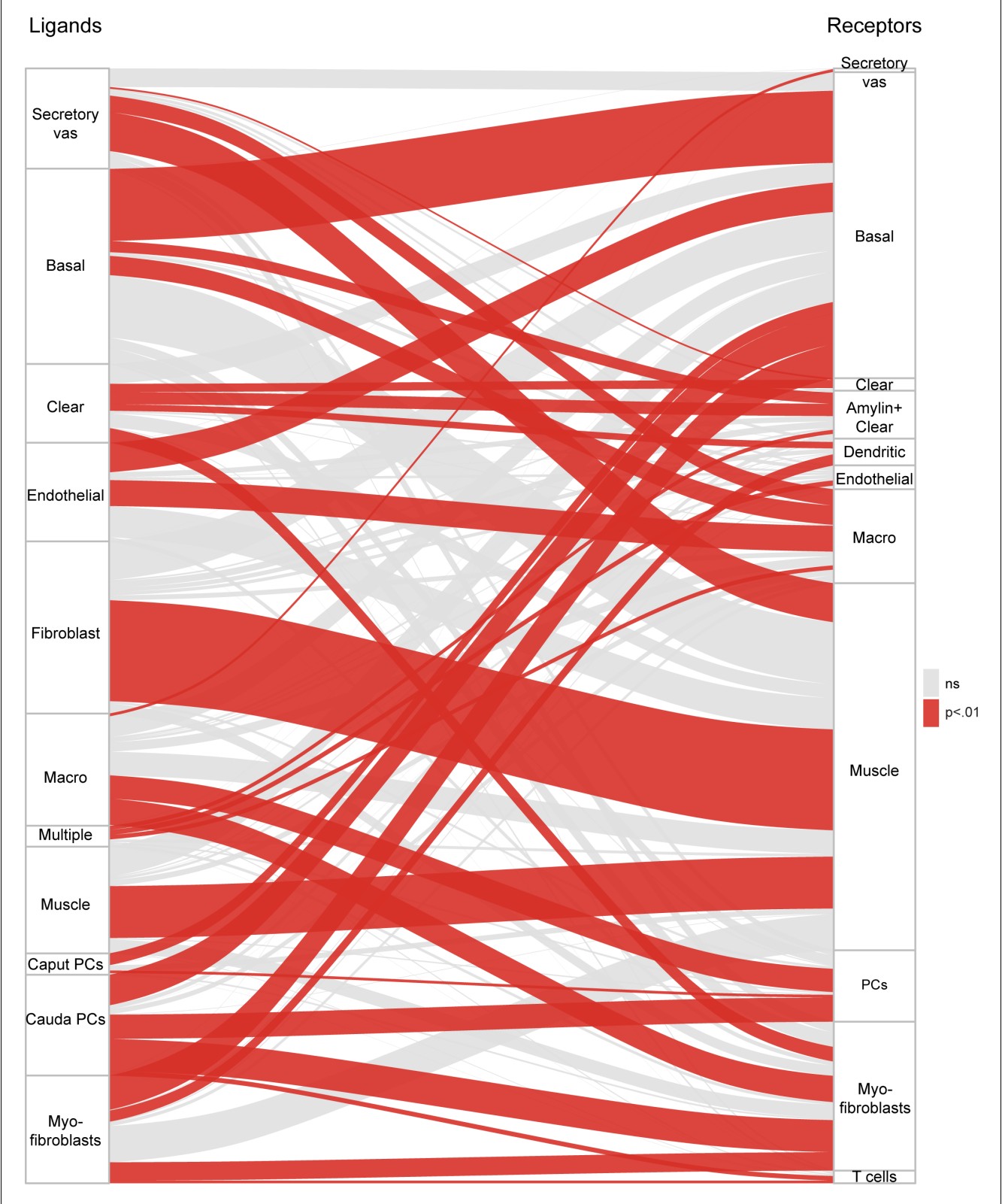

**Figure 7.** Potential signaling interactions between epididymal cell types. Bars on left and right show clusters of ligands and receptors (see *Supplementary file 4*) named according to the cell population with highest expression levels. For brevity, principal cells were abbreviated as 'PC'. Rectangle lengths represent the number of putative connections between ligands (left panels) and receptors (right panels) expressed in different cell types. Lines connecting rectangles represent connections between matching ligand-receptor pairs expressed. If the number of interactions between a

*Figure 7 continued on next page*

Figure 7 continued

cell type is equal or less than the number expected by chance (based on the number of expressed ligands and receptors in the pair of cell types, hypergeometric test), lines are grey, while lines are colored red for cell type pairs with a significant enrichment of matching ligand-receptor pairs. See *Supplementary file 4* for ligand and receptor expression in the 34 cell populations.

program with peak expression of different genes occurring transiently and asynchronously, will require long-term imaging studies using live cell reporters to address.

Turning to marker genes that are penetrantly expressed in specific regional domains, we note that a wide range of functions are represented, including signaling peptides (*Npy*, segments 7–8), putative RNA modifying enzymes (*Mettl7a1*, segment 2), and various metabolic enzymes. However, the most notable feature of region-specific gene regulation in the epididymis is the segmental expression of individual members of several large multi-gene families that are organized in genomic clusters, including β-defensins, aldo-keto reductases, lipocalins, serine protease inhibitors, and RNases (*Figure 3—figure supplement 2*). The activities encoded by these gene clusters encompass many well-characterized functions of the epididymis, as for example it has long been known that the sperm membrane and glycocalyx are extensively remodeled in the epididymis (*Bernal et al., 1980*; *Tecle and Gagneux, 2015*; *Yanagimachi et al., 1972*), while the wide array of antimicrobial peptides presumably serve to insulate the testis from ascending infections (*Dorin and Barratt, 2014*; *Gregory and Cyr, 2014*; *Hall et al., 2007*; *Ribeiro et al., 2016*). Sperm also undergo elaborate redox remodeling during their time in the epididymis, with extensive cysteine oxidation resulting in disulfide crosslinks in the genome-packaging protamines and many other proteins (*Calvin and Bedford, 1971*; *Dias et al., 2014*; *Ijiri et al., 2014*; *Saowaros and Panyim, 1979*). Conversely, another major redox function of the epididymis is to protect sperm from oxidative damage, which can cause lipid peroxidation, genome damage, and loss of sperm function (*Aitken, 2020*; *Delbès et al., 2010*; *Jones and Mann, 1977*; *Selvaratnam et al., 2015*). Consistent with these and other major redox functions of this tissue, many families of redox enzymes exhibit region-specific expression in the epididymis, as for example different members of the glutathione S-transferase-encoding gene family exhibit strong regional biases in expression (*Andonian and Hermo, 1999*; *Hales et al., 1980*; *Papp et al., 1995*). Aldo-keto reductase genes also exhibit strongly region-specific expression patterns; members of this gene family are involved in the biosynthesis of signaling molecules (*Volat et al., 2012*) such as prostaglandins (*Kabututu et al., 2009*), which have the potential to mediate both lumicrine signaling within the male reproductive tract as well as inter-partner signaling following delivery to the female reproductive tract. The aldo-keto reductases may also protect sperm from oxidative damage during transfer outside the body.

The lipocalins are a family of small molecule binding proteins which bind to lipophilic cargoes with a range of functions from signaling (retinoic acid, [*Lareyre et al., 1998*; *Ong et al., 2000*]) to innate immunity (siderophores, [*Golonka et al., 2019*]). Serine protease inhibitors are homologous to the semen coagulum proteins (which are synthesized primarily by the seminal vesicle) and potentially play some role in the protease-controlled processes of semen coagulation and liquefaction (*Clauss et al., 2011*), and may also be involved in acrosome maturation and sperm capacitation (*Ma et al., 2013*). Finally, high level expression of various RNases presumably explains the high levels of tRNA cleavage in this tissue (*Sharma et al., 2016*), with resulting tRNA fragments potentially acting as another layer of protection from selfish elements (*Martinez et al., 2017*; *Schorn et al., 2017*), or serving as environmentally modulated signaling molecules delivered by sperm to the zygote upon fertilization (*Chen et al., 2016*; *Rompala et al., 2018*; *Sarker et al., 2019*; *Sharma et al., 2016*).

Segmentation of long tubes into distinctive chemical and gene expression microenvironments is a common feature of biological tubes, ranging from the *C. elegans* gonad to the mammalian intestine. Among these, the gene expression gradients observed in the epididymis are in many ways most similar to those observed in the nephrons and collecting ducts of the kidney, another set of tubes whose function involves intricate manipulation of the pH and other chemical features of the luminal contents. Indeed, a number of genes exhibit prominent region-selective expression in both the epididymis and in ureteric principal cells, including genes of the *Aqp*, *Fxyd*, *Cry*, *Akr*, and *Tspan* families (*Ransick et al., 2019*). Other similarities in cell composition between the epididymis and kidney include V-ATPase-positive intercalated cells of the kidney, analogous to epididymal clear cells, and

the basal cell-like Krt5-positive cells of the kidney's deep medullary epithelium. That said, the functions of the epididymis and kidney also differ in many important ways, and accordingly identification of epididymis-specific genes (eg *Rnase9-12*, *Cst* family members) can be used to infer genes with functions in sperm maturation per se.

While the organization of gene expression domains in nephrons and collecting ducts plays well-understood roles in urine concentration and pH manipulation, the functions of the consecutive microenvironments of the epididymis in sperm maturation are less clear. In other words, is there any utility to having sperm first incubate for days in the presence of Defb21, then move into an incubation with Defb48, etc.? Would sperm function any differently if the various defensins (or Rnases, etc.) switched positions? The segment-specific expression of individual members of various gene families is also of mechanistic interest — what transcription factor or post-transcriptional regulator (*Björkgren et al., 2012*) is responsible for, say, expression of *Defb21* but not *Defb2* in the caput epididymis? We note that there does not appear to be any clear genomic organization relating spatial expression with genomic location within a cluster, as most famously seen in the *Hox* clusters.

## A novel subclass of clear cells

A common goal of single-cell RNA-Seq of multicellular tissues is to identify novel cell types, particularly rare cell types that might have been missed in classic histological studies. Here, we identified several cell clusters with the potential to be previously unknown subgroups of various epididymal cell types. We followed up on a subdivision of clear cells, where a subgroup of cells confined to the caput and corpus epididymis could be distinguished by expression of *Iapp* (*Figure 4*). We confirmed the presence of both Amylin-negative and Amylin-positive clear cells histologically, and found that these subgroups could even be found cohabitating within the same region of the epididymis.

The enrichment of Amylin-positive clear cells in the caput epididymis was notable given the history of morphologically distinctive clear cells, known as 'narrow' and 'apical' cells, which are both found in the initial segment of the caput epididymis (*Abou-Haïla and Fain-Maurel, 1984*; *Adamali and Hermo, 1996*; *Breton et al., 2016*; *Sun and Flickinger, 1979*; *Sun and Flickinger, 1980*). However, our Amylin-positive clear cell subset is unlikely to correspond to either of these cell types, as apical and narrow cells are confined to the initial segment of the caput epididymis, while Amylin-positive clear cells are found in downstream segments of the caput and corpus epididymis (*Figure 4—figure supplement 1C*). Moreover, one of the few reported markers of narrow cells, Carbonic Anhydrase II (*Adamali and Hermo, 1996*), is not enriched in the cells comprising Cluster two in our dataset (*Figure 4—figure supplement 1B*). Together, our data suggest that Amylin-positive clear cells do not correspond to previously identified initial segment-specific clear cells, and Amylin expression therefore may distinguish between two distinct subsets of clear cells. Whatever the correspondence to narrow cells, it will be important in future studies to determine whether functional differences can be identified that distinguish Amylin-positive and -negative clear cells – Amylin has been implicated in systemic control of metabolism (*Boyle and Le Foll, 2019*), so it will be interesting to determine whether clear cell-secreted Amylin is used to signal locally within the male reproductive tract, or whether systemic release of Amylin is used to modulate organismal metabolism, potentially in response to reproductive function.

## Perspective

Together, our data provide a bird's eye view of the cellular composition of the mouse epididymis. We identified several novel features of epididymal cell biology, most notably including a detailed examination of the understudied vas deferens. Potentially novel cell types identified include Amylin-positive clear cells, as well as two subtypes of fibroblasts – secretory and stem-like – confined to the vas deferens. Moreover, our data highlight several groups of cells with the potential for extensive intercellular signaling, and suggest that secretory fibroblasts in the vas deferens may play roles in recruiting macrophages. These data will provide a rich resource for hypothesis generation and future efforts to more fully understand the biology of the epididymis and vas deferens.

## Materials and methods

### Mice

Tissues were obtained from 10- to 12-week-old male FVB/NJ mice. All animal care and use procedures were in accordance with guidelines of the University of Massachusetts Medical School Institutional Animal Care and Use Committee (Protocol # A-1833–18).

### Dissection

In order to obtain a sperm-depleted single-cell suspension, eight epididymides from four 10- to 12-week-old FVB mice, euthanized according to IACUC protocol, were dissected into a 10-cm plate containing 10 mL of Krebs media pre-warmed to 35˚C. Using a dissection microscope with warm stage set to 35˚C, the organ was cleared of any fat and excessive connective tissue. In a clean 10-cm dish containing 10 mL of Krebs media the organ was cut into four segments that roughly corresponded to caput, corpus, cauda, and vas deferens (*Figure 1A*). Each group of eight segments was placed into different 5-cm petri dishes with about 2 mL of Krebs media. Using curved scissors, each group was further cut into roughly 1–5 mm pieces and transferred to a 15-mL conical tube. Media was added to a final volume of 10 mL, and samples allowed to decant while in an upright position at a 35˚C incubator for 3–5 min. Supernatant was discarded and this wash step repeated for two more times. In order to further remove sperm, the tubules were transferred to a 50-mL conical tube containing 25 mL pre-warmed RPMI-1640 media. After ten minutes incubation at 35˚C under mild agitation, samples were allowed to settle for 3–5 min and supernatant discarded. These steps were repeated until supernatant looked clear.

### Dissociation

Tissue dissociation was optimized to obtain a single cell suspension free of sperm. Once dissected samples were cleared of any visible sperm, 3 mL of the RPMI-1640 sample mixture was transferred to a small 25 mL glass Erlenmeyer flask containing 7 mL of dissociation media (Collagenase IV 4 mg/mL, DNAse I 0.05 mg/mL) and placed in a 35˚C water bath with rotations of 200 rpm for 30 to 45 min. Afterwards, samples were transferred to a conical tube and allowed to settle. Supernatant was removed, leaving 4–5 mL of sample solution to be returned to the Erlenmeyer flask, where 10 mL of 0.25% trypsin-EDTA and 0.05 mg/mL DNAse I was added prior to the return to the water bath shaker. After 35 min, the samples were pipetted up and down until there were no observable pieces of tissue and allowed to incubate in the water bath for an additional 20 min. Once cell disaggregation was achieved 1 mL of FBS was added to inactivate the trypsin. Sample was transferred to falcon tube while passing the cell solution through a series of 100 μm, 70 μm, and 40 μm cell strainers, and adding media (RPMI-1640 supplemented with 10% FBS and antibiotic and anti-mycotic - 100 units/mL of penicillin, 100 μg/mL of streptomycin, and 0.025 μg/mL of Amphotericin B) to a final volume of 30 mL. Finally samples were submitted to centrifugation for 15 min at 400 rcf (g) at room temperature (RT). Supernatant was discarded and up to 25 mL of media added prior to centrifugation for 5 min at 800 rcf (g) at RT. Afterwards supernatant was discarded and 2 mL of PBS added, and the cells transferred to a 2-mL microfuge tube. Media was further removed by centrifugation at 900rcf (g) for 3 min, supernatant discarded and pellet resuspended in 600 μL PBS. A 10 μL aliquot of cells plus a cell viability marker (tryphan blue), was used to determine cell viability and concentration using an automated cell counter (Bio-RAD TC20). Cells were maintained on ice and samples deemed good (>85% survival) proceeded to the downstream preparation with the 10X protocol/dilutions.

### Library protocol

Single-cell sequencing libraries were prepared using Chromium Single Cell 3' Reagent Kit V2 (10X Genomics), as per the manual, and sequenced on NextSeq 500 and HiSeq 4000 at the UMass Medical School Deep Sequencing Core.

### Data processing

Sequencing reads were processed as previously described (*Derr et al., 2016*), using the graphical interface DolphinNext [**doi:** https://doi.org/10.1101/689539]. Scripts and the full pipeline can be accessed through GitHub (github.com/garber-lab/inDrop_Processing; copy archived at https://

github.com/elifesciences-publications/inDrop_Processing; *Donnard et al., 2020*). Briefly, fastq files were generated using bcl2fastq and default parameters. Valid reads were extracted using umitools [doi:10.1101/gr.209601.116] and valid barcode indices supplied by 10X Genomics. Reads where the UMI contains a base assigned as N were discarded. Remaining reads were aligned to the mm10 genome using HISAT2 v2.0.4 (*Kim et al., 2019*) with default parameters and the reference transcriptome RefSeq v69. Alignment files were filtered to contain only reads from cell barcodes with at least 100 aligned reads and were submitted through ESAT (github.com/garber-lab/ESAT) for gene-level quantification of unique molecule identifiers (UMIs) with parameters -wLen 100 -wOlap 50 -wExt 1000 -sigTest. 01 -multimap ignore -scPrep. Finally, we identified and corrected for UMIs that were likely a result of sequencing errors, by merging the UMIs observed only once that display hamming distance of 1 from a UMI detected by two or more aligned reads.

Barcodes with at least 350 and no more than 15,000 unique molecular identifiers (UMIs) mapped to known genes were considered valid cells. Additionally, background RNA contamination was removed from the gene expression matrix using a modified version of the algorithm SoupX v0.3.1 [https://doi.org/10.1101/303727]. Briefly, barcodes corresponding to empty droplets were selected per sample according to their distribution of UMIs per barcode (caput: 20 =< x =< 175, corpus: 20 =< x =< 150, cauda: 20 =< x =< 150, vas: 20 =< x =< 115; where x = total UMIs for a given barcode). UMI counts for empty droplets were used as input for the SoupX algorithm to infer the expression profile from the ambient contamination per sample. The main alteration consisted in the selection of the fraction of contamination per cell (rho), which was determined by the most frequent number of UMIs for the empty droplets in each sample (caput = 139; corpus = 114; cauda = 109; vas = 84), under the assumption that the same contamination level could be expected from valid cell containing droplets. Using these estimated contamination fractions, the SoupX algorithm was used to remove counts from genes that had a high probability of resulting from ambient RNA contamination for each cell using the function adjustCounts.

## Dimensionality reduction and clustering

Gene expression matrices for all samples were loaded into R (V3.6.0) and merged. The functions used in our custom analysis pipeline are available through an R package (github.com/garber-lab/SignallingSingleCell). Using the clean raw expression matrix, the top 20% of genes with the highest coefficient of variation were selected for dimensionality reduction. Dimensionality reduction was performed in two steps, first with a PCA using the most variable genes and the R package irlba v2.3.3, then using the first 7 PCs (>85% of the variance explained) as input to tSNE (Rtsne v0.15) with parameters perplexity = 30, check_duplicates = F, pca = F (*van der Maaten and Hinton, 2008*). Clusters were defined on the resulting tSNE 2D embedding, using the density peak algorithm (*Rodriguez and Laio, 2014*) implemented in densityClust v0.3 and selecting the top 30 cluster centers based on the γ value distribution (γ = ρ × δ; ρ = local density; δ = distance from points of higher density). Using known cell type markers, clusters were used to define six broad populations (Immune, Stromal, Basal, Principal, Clear and Muscle). UMI counts were normalized using the function computeSumFactors from the package scran v1.12.1 (*Lun et al., 2016*), and parameter min. mean was set to select only the top 20% expressed genes to estimate size factors, using the six broad populations defined above as input to the parameter clusters. After normalization, only cells with size factors that differed from the mean by less than one order of magnitude were kept for further analysis $(0.1 \times (\Sigma(\theta/N))) > \theta > 10 \times (\Sigma(\theta/N)); \theta$ = cell size factor; N = number of cells). Normalized UMI counts were converted to gene fractions per cell and the resulting matrix was used in a second round of dimensionality reduction and clustering. The top 15% variable genes were selected as described above and used as input to PCA and the first 24 PCs that explained >95% of the variance were used as input to tSNE. The resulting 2D embedding was used to determine 21 clusters as described above. Marker genes for each cluster were identified by a differential expression analysis between each cluster and all other cells, using edgeR (*McCarthy et al., 2012*), with size factors estimated by scran and including the sample of origin as the batch information in the design model. Each of the six broad cell type populations was independently re-clustered following the same procedure described above, to reveal more specific cell types. Expression values per cluster were calculated by aggregating gene expression values from individual cells in each cluster and normalizing by the total number of UMIs detected for all cells in that cluster, multiplied by $10^6$ (UMIs per million). In scRNA-Seq data, zero values for a given gene can result from technical dropouts or from true

biological variability where cells do not express that gene. To estimate the fraction of true zeros for each gene, we used a zero inflated negative binomial model implemented in DESingle (*Miao et al., 2018*). From this model we calculated the fraction of cells in any given cluster expressing each gene displayed in *Figure 3—figure supplement 1*, with outputs in *Supplementary file 3* including the fraction of expressing cells as well as expression level only in positive cells.

An R script for the steps described here and containing all parameters used is available through GitHub (github.com/elisadonnard/SCepididymis; copy archived at https://github.com/elifesciences-publications/SCepididymis; *Donnard, 2020*).

## Receptor-ligand network analysis

Receptors and ligands expressed in the dataset were selected based on a published database (*Ramilowski et al., 2015*). The database genes were converted to their respective mouse homologs using the Homologene release 68. Genes were filtered to select only those expressed by at least 20% of cells in at least one cell population, resulting in a set of 226 ligands and corresponding 212 receptors. Aggregated expression values were calculated based on this filtered gene matrix per cell type as described above. The annotation of each pair of expressed ligands and receptors in the data was done using the function *id_rl* from the package *SignallingSingleCell*. The expression matrix for all annotated ligands or receptors was clustered using *k-means* (k = 13 ligands; k = 14 receptors) and clusters were labeled and merged based on the cell population showing highest expression of those genes, resulting in 11 final ligand cell groups and 11 final receptor cell groups (*Supplementary file 4*). The frequency of connections from each ligand to each receptor cluster was calculated and significant cell-cell interactions (p<0.01) were determined with a hypergeometric test (*Figure 7*).

## Histology

Male mice from the same strain and age as previously described were anesthetized and perfused with a solution of phosphate-buffered saline (PBS) followed by 4% paraformaldehyde (PFA)/PBS, and epididymides explanted and further incubated in fixative at 4°C overnight (ON). After washing the excess of PFA with PBS, the sample was incubated at 4°C with a 30% sucrose, 0.002% sodium azide (NaAz) in PBS solution for 32 hr, after which a similar volume of optimal cutting temperature compound (OCT) was added to the vial and kept ON at 4°C under agitation. Samples were than mounted using OCT and frozen at −80°C until sectioning. Sectioning was done at 5 μm thickness by the UMASS morphology core. Slides were stored frozen at −80°C until used for immunofluorescence.

## Immunofluorescence

Slides were placed at a 37°C warm plate for at least 30 min to ensure proper attachment of the section to the slide, then washed three times of 5 min in PBS 0.02% tween 20 (PBS-T) to remove OCT. Staining was performed as suggested by the anti-body manufacturer. Briefly a 10 min permeabilization step in PBS 0.1% triton-X was performed prior to a 2 hr blocking with a PBS blocking solution containing 5% goat serum, 3% BSA and 3%goat serum, or following the Vector Laboratories' Mouse on Mouse detection kit (M.O.M.). The primary and secondary antibodies specifications and dilutions are in *Supplementary file 5*. All slides were incubated with primary antibodies at 4°C ON, washed three times of 5 min with PBS-T and subsequently incubated with Alexa Fluor secondary antibodies for 3 hr followed by a 5 min wash with PBS-T, than by DAPI stain for 5 min, followed by three washes of at least minutes with PBS_T. Slides were mounted using ProLong gold Antifade (Thermofisher) and imaged the following day using a Zeiss Axio inverted microscope. Images were then processed to increase brightness and contrast using Photoshop.

## Acknowledgements

We thank the Luban and Greer labs for providing instruments and guidance with single-cell sequencing, and members of the Rando and Garber labs for critical reading of the manuscript and insightful discussions. This work was supported by Templeton Foundation grant 61350 and NIH grant R01HD080224. US is supported by NIH grant 1DP2AG066622-01 and Searle Scholars Program.

## Additional information

### Funding

| Funder | Grant reference number | Author |
|---|---|---|
| Eunice Kennedy Shriver National Institute of Child Health and Human Development | R01HD080224 | Vera D Rinaldi<br>Elisa Donnard<br>Kyle Gellatly<br>Morten Rasmussen<br>Alper Kucukural<br>Onur Yukselen<br>Manuel Garber<br>Oliver J Rando |
| NIH Office of the Director | 1DP2AG066622 | Upasna Sharma |
| John Templeton Foundation | 61350 | Vera D Rinaldi<br>Oliver J Rando |

The funders had no role in study design, data collection and interpretation, or the decision to submit the work for publication.

### Author contributions

Vera D Rinaldi, Conceptualization, Data curation, Formal analysis, Investigation, Visualization, Writing - original draft, Writing - review and editing; Elisa Donnard, Data curation, Software, Formal analysis, Visualization, Writing - review and editing; Kyle Gellatly, Methodology; Morten Rasmussen, Investigation; Alper Kucukural, Onur Yukselen, Resources; Manuel Garber, Data curation, Formal analysis, Supervision, Methodology; Upasna Sharma, Conceptualization, Data curation, Validation, Investigation; Oliver J Rando, Conceptualization, Supervision, Funding acquisition, Visualization, Writing - original draft, Project administration, Writing - review and editing

### Author ORCIDs

Vera D Rinaldi (iD) http://orcid.org/0000-0002-0051-1754
Elisa Donnard (iD) http://orcid.org/0000-0002-8834-8110
Alper Kucukural (iD) http://orcid.org/0000-0001-9983-394X
Manuel Garber (iD) http://orcid.org/0000-0001-8732-1293
Oliver J Rando (iD) http://orcid.org/0000-0003-1516-9397

### Ethics

Animal experimentation: All animal care and use procedures were in accordance with guidelines of the University of Massachusetts Medical School Institutional Animal Care and Use Committee (Protocol # A-1833-18).

### Decision letter and Author response

Decision letter https://doi.org/10.7554/eLife.55474.sa1
Author response https://doi.org/10.7554/eLife.55474.sa2

## Additional files

### Supplementary files

• Supplementary file 1. Bulk RNA-Seq dataset. RNA-Seq for caput (CP), corpus (CO), and cauda (CA) epididymis, and for vas deferens (VD). All data are normalized to parts per million.

• Supplementary file 2. Marker gene expression across 21 cell clusters. Log2 fold enrichment for 11089 genes across the 21 cell clusters (*Figure 2A*) from the full dataset.

• Supplementary file 3. Principal cell subclustering and cell to cell heterogeneity. Output of DESingle for the 15 principal cell subclusters (*Figure 3B*). For each subcluster, table includes fraction of cells not expressing the gene in question ('theta_2', or estimated true zeroes), average expression across

all cells ('cluster_UPM', expressed as UMIs per million), and average expression level only in positive cells ('expressing_cell_meanUPM').

• Supplementary file 4. Ligand and receptor expression across epididymal cell types. Relative expression levels for all ligands or receptors expressed (>=1 UMI) in at least 20% of cells from one or more of the 34 different cell types listed.

• Supplementary file 5. Antibodies used for immunofluorescence studies. List of antibodies used in IF studies throughout the manuscript.

• Transparent reporting form

## Data availability

Data are available at GEO, Accession #GSE145443.

The following dataset was generated:

| Author(s) | Year | Dataset title | Dataset URL | Database and Identifier |
|---|---|---|---|---|
| Rinaldi VD, Donnard E, Gellatly KJ, Rasmussen M, Kucukural A, Yukselen O, Garber M, Sharma U, Rando OJ | 2020 | An atlas of cell types in the mammalian epididymis and vas deferens [single-cell RNA-Seq] | https://www.ncbi.nlm.nih.gov/geo/query/acc.cgi?acc=GSE145443 | NCBI Gene Expression Omnibus, GSE145443 |

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
