## [Decision Letter]

**Acceptance summary:**

Using microfluidic-based cell isolation and molecular barcoding strategies coupled with genome-wide deep sequencing technologies, this study provides an important reference that re-examines gene expression profiles of all cell types along the epididymis and vas deferens of the mouse. These comprehensive results reinforce the previously established remarkable changes in gene expression that occur as one traverses this long duct. The clustering of genes to the different cell populations in the epididymis and vas deferens (within a single study), the inclusion of the vas deferens in this analysis, and the identification of putative new subpopulations of clear and basal cells should make this study a valuable resource to all researchers in this field.

**Decision letter after peer review:**

Thank you for submitting your article "An atlas of cell types in the mammalian epididymis and vas deferens" for consideration by *eLife*. Your article has been reviewed by four peer reviewers, and the evaluation has been overseen by a Reviewing Editor and Didier Stainier as the Senior Editor. The following individuals involved in review of your submission have agreed to reveal their identity: Gail Cornwall (Reviewer #2); Louis Hermo (Reviewer #3).

The reviewers have discussed the reviews with one another and the Reviewing Editor has drafted this decision to help you prepare a revised submission.

Summary:

This manuscript is timely and of considerable interest to researchers working in the field of the epididymis. It is a re-examination of mouse epididymis gene profiling using microfluidic-based cell isolation and molecular barcoding strategies coupled with genome-wide deep sequencing technologies. The purpose is to dissect out the contributions of different cell types to the highly regionalized gene expression profiles in the mouse epididymis and vas deferens. Although most of the authors' conclusions, including that distinct cell populations such as that unique subsets of principle cells express different genes which leads to changing luminal microenvironments along the tubule, are not at all new to the field, information is presented that does provide new "tools". This includes the clustering of genes to the different cell populations in the epididymis and vas deferens (within a single study), the inclusion of the vas deferens in this analysis, and the identification of putative new subpopulations of clear and basal cells.

While this study takes a novel approach to understand the diversity of cell types of the epididymis and vas deferens, there are a number of major points that need to be addressed.

Essential revisions:

1) The title should be "mouse" not "mammalian"- it is well known that there are profound species differences in genital duct structure and function.

2) Introduction. The Introduction has an inadequate bibliography. For example, only one paper from 2007 specifically related to studies on the transcriptome of the human epididymis is cited. Several others relevant papers from different research groups, both from the mid 2000's and more recently are ignored.

3) Materials and methods. Unfortunately, the bioinformatics-based predictions of regional distribution of subclasses of key cell types (e.g., principal cells) and new cell types (e.g., variant clear cells) presented here are not been validated by another experimental protocol, preferably immunohistochemistry. The PI has the tissues and the skill set required to produce these imaging data and is encouraged to provide them in a revised manuscript.

4) Materials and methods. The choice to divide the mouse epididymis into three sections as the basis for the rest of the studies presents a major problem, given what is already well established in the literature about this tissue. It has long been known that the gross separation caput/corpus/cauda reflects neither the finely segmented anatomy of the organ nor its functional segmentation. The studies in this manuscript could have provided a major enhancement on the excellent studies done by Johnston et al., 2005, where anatomical segmentation of the mouse epididymis into 10 segments was described. Although it is understandable that separation into 10 segments would have been challenging, separating the segments by following at least the 5 functional domains previously identified by Johnston's work (i.e. S1, S2, S3 to S6, S7, S8 to S10) would have been appropriate. Such a functionally more appropriate approach is much more powerful with respect to this segmented organ, and how this may affect its physiology and cell-cell interactions. As an example of strange differences between the two studies, how is it that *Gpx5*, which is undoubtedly a highly expressed caput gene (representing about 5% of the total caput transcripts), does not appear in Figure 1—figure supplement 1? The same applies to *Gpx3* in the cauda epididymidis. These are just a few examples. Please explain why that was not done.

Further, even if one accepts the segments used for this study, the authors make no effort to use the scRNAseq data to attempt to functionally separate cells of the initial segment from other caput cells. Since some species lack an initial segment, it will be useful to compare the gene expression profile from the initial segment with, for example, efferent ducts in the human epididymis.

5) Figure 1. The diagram of the mouse genital duct in Figure 1A is unacceptable. Please show an image of real tissue and the septa used to define the dissection. One of the advantages of mouse is the ability to do precise dissections according to location of septa in the epididymis.

6) Figure 2. The single-cell approach presented in Figure 2 is interesting because it allows localization of the expression of particular genes in cell subtypes. In panel 2C, what was the rationale for illustrating the selected genes and not systematically the genes that are considered markers of epididymal sub-cell types and that are listed in the right margin of Figure 2A? It is also difficult to understand the C8, C18, C11 distinction in the principal cells; C6 and C11 being caudal principal cells whereas C18 comes from the corpus. What are the reasons for this and why is the numbering not logical from caput to caudal?

The data suggesting regional gene expression profiles of different principal cells along the epididymis need to be confirmed by immunofluorescence or other histological methods. Other explanations could underlie these cell clusters.

7) Similarly, the data in Figure 3 that shows genomic clusters of expressed genes from multigene families, needs to be validated by direct visualization of marker proteins.

8) Figure 3—figure supplement 3, localization of Abcg2 in the vas deferens requires an image of tissue incubated with a control serum (normal IgG, or better blocked antibody with peptide; secondary antibody alone is not an appropriate control for IIF). This could replace one of the panels shown (which are just two images at two different magnifications); a scale bar is also necessary.

9) Figure 4E is not adequate as there is no orientation of the tissue section and no illustration of negative controls or other markers.

10) The identification of two types of clear cells by re-clustering is novel. The use of amylin as a marker of the second group is also of interest, but it would be important to know if this marker is also found in other clusters of epididymis cells before arriving at this interpretation. There is the speculation that the amylin positive cells are "narrow cells" but again more extensive validation using imaging is required to support this conclusion.

Further, the finding that different populations of a cell type, even within a single cross-section, can have different levels of gene expression needs to be considered and properly referenced. This has also been noted for principal cells and their expression of many different proteins (Hermo et al., 1991, 1992; Rankin et al., 1991), where in a given cross section some principal cells are highly reactive while adjacent principal cells were unreactive. In these cases, it was proposed that the principal cells were not in synchrony with respect to the secretion of these proteins. However, it may also suggest that the principal cell population of a given region is more diverse than ever imagined. Likewise, one could argue that not all the clear cells of a given region are not in synchrony with respect to the expression of a given protein at one given time or that the population of clear cells is more diverse than expected. Examples of diverse expression of GSTs in principal cells also highlights the hypothesis of diverse populations of principal cells in a given epididymal region (Papp et al., 1995). Please consider including these points in your data interpretation.

11) The description of a potential new population of basal cells is interesting but no indication is given as to whether these were restricted/enriched in a particular epididymal region. Please provide this information.

12) Basal cells have been suggested to be derived from the monocyte/macrophage lineage (Seiler et al., 1998, 1999). The group of Breton has indicated that basal cells are COX-1 and cytokeratin 5+ cells and distinct from the F4/80 macrophages. In addition, dendritic cells are present in the epithelium. Furthermore, all of these cells are lodged at the base of the epithelium where they firmly adhere to the basement membrane. Without the use of appropriate markers to separate these cells, how can we be certain that what is being called basal cells are not in fact a mixture of the monocyte/macrophage basal cells of the Cooper group as compared with the dendritic cells of the Breton group. LM images and quantification would appear to be a way to confirm what basal cell type was extracted.

13) Although in this study, macrophages are indicated to be strongly linked to the vas deferens, no indication of epididymal region is given for the other immune cell populations. Furthermore, in the heading of immune cells, the authors define a category of endothelial cells. There is no evidence of any endothelial cells being present in the epididymal epithelium. This suggests that the preparations are contaminated. Endothelial cells are found lining blood vessels and these do not exist in the epithelium. Can the authors clarify from where the endothelial cells were derived? As for the immune cells, once again some LM images and quantification of percent of each of the different immune cells examined would be in order, e.g. macrophages, dendritic cells, T-lymphocytes, etc).

14) The cell:cell communication aspect shown by bioinformatics comparisons in Figure 6 is interesting. However, the speculation that basal cells and endothelial cells, and macrophages and "secretory fibroblasts", in the vas express complementary sets of ligands and receptors needs to be validated by another method, e.g., immunohistochemistry, to demonstrate that these cells are co-localized. Please also define PC , DC, SI, NS.

15) Analysis of the vas deferens is of interest as this part of the duct is rarely studied. The authors note high expression of genes involved in mitochondrial energy metabolism. Is it possible that the cells isolated from the vas include spermatozoa? The tissue samples are from 10-14 weeks mice, when males are already sexually mature. Though the single cell isolation protocol stresses the removal of sperm from tissue, it appears to rely on visual inspection of supernatant turbidity. Some other markers should be investigated to exclude the possibility of contaminating sperm.

16) While the fluorescent images are good, they do not cover all regions of the duct, and there is no way of knowing if the images are representative. How many animals were used and how many experiments were done to ensure reproducibility?

17) The Discussion is remarkably incomplete and somewhat biased. Some specific questions that are not satisfactorily addressed are:

1) To which cell type do the so-called apical cells belong? Apical cells are not discussed at all in this manuscript;

2)- The discussion on immune cells is poor considering the recent publications on murine epididymal immune cells from several laboratories. Please refer to and discuss the studies focussed on epididymal immune cells, e.g., Voisin et al., 2018;

3) There is a complete lack of discussion of the major and well-known roles of epididymal sperm maturation concerning the fine balance between oxidative and anti-oxidative processes. These are just a few examples, as there are many landmark manuscripts dealing with epididymal physiology that have been ignored.

In addition, the speculation in the Discussion on the highly segmental expression of several large multigene families and its relative importance is "epididymocentric". It could be argued that this is a feature of any tubular structure in the body and the authors are encouraged to examine (for example) the single cell data from the mouse kidney before making the assumption that this is an epididymis-specific feature.

---

## [Author Response]

We thank the reviewers for the detailed and constructive comments. In addressing these comments, we have learned a great deal about the history of epididymal gene regulation, and we believe we have significantly improved the utility of the manuscript to the community.

The major new data included in this revised manuscript, based on quite a few distinct requests from the reviewers, are a number of immunostaining studies. Specifically, we now include new images for the following studies:

– ATP6V1G3 (clear cell marker) + Amylin. We had already included some images of these data, described a subset of Amylin-positive clear cells, in the original manuscript. As requested by reviewers, we have added additional images of this staining effort from different parts of the epididymis, confirming our claim that the Amylin-positive clear cells are confined to the caput and early corpus (**Figure 4 and Figure 4—figure supplement 1C**) as well as validating our findings using independent antibodies.

– KRT5 (basal cells) + F4/80 (Macrophages and monocytes), KRT5 + MHC Class II (antigen-presenting cells). These new staining data address reviewer comments regarding the distinction, or potential lack thereof, between basal cells and various mononuclear phagocytes. We confirm prior reports documenting clear distinctions between basal cells and macrophages and other APCs (**Figure 5—figure supplement 1C-D**).

– F4/80 and MHC Class II images in the vas deferens, responding to the reviewer concern that these APCs might only be found in contaminating blood vessels, and demonstrating the presence of interstitial APCs (**Figure 5—figure supplement 1C**).

– Vas deferens staining with Lgr5 and either Ptx3 or Tslp, confirming our distinction between two stromal cell types (“secretory” and “stem-like”) in the vas deferens (**Figure 6—figure supplement 2**).

The other major addition to the manuscript is an explicit analysis of cell-to-cell heterogeneity in gene expression within principal cell subclusters, presented in the new **Figure 3—figure supplement 1** and **Supplementary file 3**. This analysis proved quite interesting, as we find evidence – consistent with the prior staining studies from Hermo, Rankin, Papp highlighted by the reviewers – for different genes to be expressed either more penetrantly or more variably. For example, we confirm that Clusterin is unusually “patchy” relative to other genes with the same expression level, consistent with staining studies. We think this analysis adds substantially to the manuscript and thank the reviewers for suggesting the idea.

Beyond these, other major changes to the manuscript include reanalysis of the relationship between basal cells and mononuclear phagocytes, new or more extensive discussions of several requested topics, including increasing citations to prior literature, and a discussion comparing epididymal gene regulation to that observed in similar organs like the kidney. Together, we believe these changes have greatly increased the value of this resource to the community.

Essential revisions:1) The title should be "mouse" not "mammalian"- it is well known that there are profound species differences in genital duct structure and function.

Fixed as requested.

2) Introduction. The Introduction has an inadequate bibliography. For example, only one paper from 2007 specifically related to studies on the transcriptome of the human epididymis is cited. Several others relevant papers from different research groups, both from the mid 2000's and more recently are ignored.

We have added the following references on other genomic studies on the epididymis: Dube et al., 2007, Zhang et al., 2006, Browne et al., 2019, Legare and Sullivan, 2019. We have also expanded our citations on several other topics (immune cell populations, comparison to the kidney, cell to cell heterogeneity, etc.) as detailed in response to the appropriate review points.

3) Materials and methods. Unfortunately, the bioinformatics-based predictions of regional distribution of subclasses of key cell types (e.g., principal cells) and new cell types (e.g., variant clear cells) presented here are not been validated by another experimental protocol, preferably immunohistochemistry. The PI has the tissues and the skill set required to produce these imaging data and is encouraged to provide them in a revised manuscript.

This is one of many comments requesting immunostaining-based validation, which we address in the introduction to the revision. With respect to the specific issues raised here, the reviewer is partly mistaken – our initial manuscript DID include IF studies validating the presence of Amylin-positive clear cells in the caput. Beyond that, carrying out additional lab experiments was challenging due to the CoVid pandemic, and among the various specific follow-up efforts, validation of principal cell gene expression patterns was quite low on our priority list for two major reasons: 1) our data are so strongly concordant with many prior studies; 2) many of the region-specific gene expression differences are seen in large multi-gene families, where high quality antibodies that distinguish family members are scarce (if they exist at all).

We do have one image that could be relevant, a light sheet image of Defb41 Cre-driven tdTomato expression in the caput epididymis. But the Defb41 Cre has previously been validated as caput-specific, so we do not feel this adds much.

4) Materials and methods. The choice to divide the mouse epididymis into three sections as the basis for the rest of the studies presents a major problem, given what is already well established in the literature about this tissue. It has long been known that the gross separation caput/corpus/cauda reflects neither the finely segmented anatomy of the organ nor its functional segmentation. The studies in this manuscript could have provided a major enhancement on the excellent studies done by Johnston et al., 2005, where anatomical segmentation of the mouse epididymis into 10 segments was described. Although it is understandable that separation into 10 segments would have been challenging, separating the segments by following at least the 5 functional domains previously identified by Johnston's work (i.e.: S1, S2, S3 to S6, S7, S8 to S10) would have been appropriate. Such a functionally more appropriate approach is much more powerful with respect to this segmented organ, and how this may affect its physiology and cell-cell interactions. As an example of strange differences between the two studies, how is it that Gpx5, which is undoubtedly a highly expressed caput gene (representing about 5% of the total caput transcripts), does not appear in Figure 1—figure supplement 1? The same applies to Gpx3 in the cauda epididymidis. These are just a few examples. Please explain why that was not done.

We agree that there are more meaningful ways to segment the epididymis than the simple caput/corpus/cauda dissection presented here. However, single cell RNA-Seq behaves counterintuitively in terms of tissue sampling: the more individual dissections done, the more substantial and complex the analysis that goes into removing contamination by cell-free RNA (see our Materials and methods).

Indeed, if anything we actually regret not simply analyzing the entire epididymis without **any** dissection – the major advantage of single-cell RNA-Seq is that no assumptions whatsoever have to be made about the boundaries of functionally-distinct territories, the data will highlight the different cell types if meaningful differences exist (as of course they do). Thus, with the exception of the bulk RNA-Seq in **Figure 1B** and **Figure 1—figure supplement 2** (originally **Figure 1—figure supplement 1**), the dissection margins do not meaningfully influence the dataset. Principal cells are easily mapped to known segments based on comparison to prior RNA-Seq studies, for example, given the clear fact that these cells contribute the major segment-defining genes in prior studies.

In the end, the only way that the dissection margins helped here was to search for region-specific expression patterns in the accessory (basal clear, etc.) cell types. We found two uses for this information – the amylin-positive clear cells in the caput, and the region-specific localization of certain immune cells. Otherwise we find very little evidence for segment or region-based specialization of accessory cell types in the epididymis. Thus, the dissection margins are immaterial for this study – there would not have been much to gain from looking specifically at basal cell density in segment 5 vs. segment 7 of the corpus.

As for the specific queries regarding *Gpx5* and *3*, we have now included *Gpx3* as requested (it was simply not listed among the genes of interest for us in the image, but it was one of the unlabeled genes in the heatmap). As for *Gpx5*, **Figure 1—figure supplement 2** (old **Figure 1—figure supplement 1**) is focused on extremely segment-restricted genes from Johnston et al., 2005, and as it happens *Gpx5* did not fulfil our criteria in the Turner, 2005 data, as it was expressed at nearly identical levels across FOUR segments: *Gpx5* is indeed highly enriched in the caput as expected but very flat across segments 1-4: 1045 / 1110 / 1272 / 1200 ppm). And our data nicely validates this, showing precisely the pattern the reviewer expects: mRNA abundances for *Gpx5* are ~39000 / 14 / 68 / 50 ppm for caput, corpus, cauda, and vas, exactly behaving as predicted. So our data perfectly confirm prior art here. We also now illustrate this in new **Figure 2—figure supplement 1A**.

More generally speaking, as far as our choices for genes to show in figures, we chose to highlight a subset of genes (rather than every gene studied in the epididymis in the literature) first of all to validate our approach, and then second of all we also included genes of interest to us specifically (RNases for instance).

Finally, as this is a Resource for interested researchers, full lists of genes can be found in Supplementary file **2**.

Further, even if one accepts the segments used for this study, the authors make no effort to use the scRNAseq data to attempt to functionally separate cells of the initial segment from other caput cells. Since some species lack an initial segment, it will be useful to compare the gene expression profile from the initial segment with, for example, efferent ducts in the human epididymis.

The initial segment corresponds to what was initially called Cluster 15 in **Figure 3**, which we now annotate (and rename as Cluster 1). This is based on maximal expression of the same genes in Cluster 15 (now Cluster 1) and in Segment 1 from Johnston et al., 2005 (*RNase10*, *Cst11*, *Lcn2*, *Mfge8*, and others).

As for comparing to human efferent ducts, we compared both our data and the Turner et al. dataset to the analysis of human efferent ducts in Legare and Sullivan, 2019. By and large the genes that are noted to be enriched in human efferent ducts in Legare and Sullivan – genes involved in cilia function (*Dnah* genes, *Ift* genes), signaling genes (*Esr1*, *Esrrg*, *Hnf4a*, etc.) and sperm-related genes (*Piwil4*, *Tsga10*, etc.) – are not enriched in our Cluster 1, nor enriched in Turner et al. Segment 1. We could note this briefly in our revised text, although we have not made any effort generally to compare to other species’ datasets (and many segment-specific genes are rapidly-evolving, making ortholog assignment a nontrivial task) so unless specifically requested we were going to skip this.

5) Figure 1. The diagram of the mouse genital duct in Figure 1A is unacceptable. Please show an image of real tissue and the septa used to define the dissection. One of the advantages of mouse is the ability to do precise dissections according to location of septa in the epididymis.

We have replaced the **Figure 1A** cartoon with an image of a typical epididymis dissection, as requested. We have also included light sheet images of mouse epididymis in new **Figure 1—figure supplement 1**, as these are compelling images that further illuminate the internal gross anatomy of the tissue for the benefit of non-experts.

6) Figure 2. The single-cell approach presented in Figure 2 is interesting because it allows localization of the expression of particular genes in cell subtypes. In panel 2C, what was the rationale for illustrating the selected genes and not systematically the genes that are considered markers of epididymal sub-cell types and that are listed in the right margin of Figure 2A?

For this figure we chose a combination of canonical marker genes and a selection of genes of interest. In some cases, the canonical marker gene is expressed at relatively low levels for single cell RNA-Seq. For example, although we DO see *Aqp9*, a commonly-used marker of Principal Cells, expressed across most of the Principal Cell clusters, it is present at relatively low abundance so the “gene paint” figures are speckled and thus less visually compelling. On the other hand, although *Spink8* is slightly less ideal in terms of its distribution (it is expressed at lower levels in caput PCs than the rest of the epididymis), it is robustly expressed and very clear as a gene paint image. The same logic leads us to use *Cldn4* rather than *Krt5* for basal cells. Finally, for some of the cell types here there is no clear canonical marker we can find in the literature.

In any case, it is important to note that the heatmap in **Figure 2B** has more room for gene names for each cluster, and since it aggregates data from all cells in a cluster it is clearer for poorly-expressed genes; most of the information one would want is present in this figure panel.

It is also difficult to understand the C8, C18, C11 distinction in the principal cells; C6 and C11 being caudal principal cells whereas C18 comes from the corpus. What are the reasons for this and why is the numbering not logical from caput to caudal?

For this figure we chose a combination of canonical marker genes and a selection of genes of interest. In some cases, the canonical marker gene is expressed at relatively low levels for single cell RNA-Seq. For example, although we DO see *Aqp9*, a commonly-used marker of Principal Cells, expressed across most of the Principal Cell clusters, it is present at relatively low abundance so the “gene paint” figures are speckled and thus less visually compelling. On the other hand, although *Spink8* is slightly less ideal in terms of its distribution (it is expressed at lower levels in caput PCs than the rest of the epididymis), it is robustly expressed and very clear as a gene paint image. The same logic leads us to use *Cldn4* rather than *Krt5* for basal cells. Finally, for some of the cell types here there is no clear canonical marker we can find in the literature.

In any case, it is important to note that the heatmap in **Figure 2B** has more room for gene names for each cluster, and since it aggregates data from all cells in a cluster it is clearer for poorly-expressed genes; most of the information one would want is present in this figure panel.

The data suggesting regional gene expression profiles of different principal cells along the epididymis need to be confirmed by immunofluorescence or other histological methods. Other explanations could underlie these cell clusters.

This is a cryptic statement. What “other explanations” could explain the marked gene expression differences between principal cell clusters, which **perfectly** overlap with MANY prior gene expression analyses of regional gene regulation in the epididymis? We would be very happy to modify our manuscript if we were informed of any legitimate alternative explanations that would explain how our data would so be perfectly consistent with scores of prior studies; we note that as far as we can tell, any such explanations would require that **the prior studies somehow *also* be marred** by the artifact in question.

Focusing just on the comparison to Johnston et al., 2005 (the dataset we used as our 10 segment “bible”) , our 11 principal cell clusters from the epididymis (not including our 4 vas deferens clusters) in **Figure 3** very cleanly recapitulate segmental gene expression patterns – based on highly-enriched marker genes, our clusters from 1 to 11 (new numbering) correspond to segment 1 (initial segment), segments 1-2, segments 3-4, segment 5 (early), late segment 5/segment 6, segment 7, late segment 7, segments 8-9, segment 9, late segment 9 (some segment 10 markers), segment 10. We now detail this in the legend for **Figure 3D**.

In addition to recapitulating **extensive** prior studies, we note that few antibodies exist for genes that distinguish region-specific principal cells, as they are typically distinguished by members of paralogous multi-gene families, many of which are secreted. Even before the CoVid pandemic stopped the majority of experimental work in our lab, we did not think it would be worthwhile to explore the complete catalog of Defb antibodies to find reasonable markers when multiple other lines of evidence support the fact that principal cells from different segments exhibit differences in gene expression.

7) Similarly, the data in Figure 3 that shows genomic clusters of expressed genes from multigene families, needs to be validated by direct visualization of marker proteins.

We have covered this exhaustively above. We do not think the requested IF is likely to succeed, and we do not see any legitimate scientific benefit to doing it either.

8) Figure 3—figure supplement 3, localization of Abcg2 in the vas deferens requires an image of tissue incubated with a control serum (normal IgG, or better blocked antibody with peptide; secondary antibody alone is not an appropriate control for IIF). This could replace one of the panels shown (which are just two images at two different magnifications); a scale bar is also necessary.

As requested, we have added a scale bar to what is now Figure 3—figure supplement 4, and we provide several negative control images throughout the manuscript stained with antibodies to proteins not expressed in a given region.

9) Figure 4E is not adequate as there is no orientation of the tissue section and no illustration of negative controls or other markers.

We are not clear as to the request for “orientation of the tissue section” – we would be happy to indicate whatever is needed, but we do not understand the request. We believe this is addressed by the images of other regions of the epididymis in the new **Figure 4—figure supplement 1C**, but if not, we are unsure what was requested. As for negative controls, here for example the new images of Abcg2 staining in the epididymis (where it is not expressed) provides a new staining control.

10) The identification of two types of clear cells by re-clustering is novel. The use of amylin as a marker of the second group is also of interest, but it would be important to know if this marker is also found in other clusters of epididymis cells before arriving at this interpretation. There is the speculation that the amylin positive cells are "narrow cells" but again more extensive validation using imaging is required to support this conclusion.

We have addressed this comment in several ways. First, new **Figure 4—figure supplement 1A-B** show *Iapp* expression across all cells in our main dataset (A) and across clear cells only (B), showing the restriction of *Iapp* to a subset of clear cells. Second, the new IF images in **Figure 4—figure supplement 1C** now include several additional sections stained for V-ATPase and Amylin.

In the original manuscript we did note that Amylin-positive clear cells might not be narrow cells, given the presence of Amylin-positive cells in both the initial segment but also in the caput and corpus. In addition, Amylin-negative clear cells in our image also show the “champagne” glass apical nucleus attributed to narrow cells. So altogether we do not believe that Amylin-positive clear cells represent narrow (nor apical) cells, but instead a distinctive subset of clear cells. We have tried to be clearer about the relationship between Amylin-positive clear cells and narrow and apical cells in the revised text describing the analysis of clear cells.

Further, the finding that different populations of a cell type, even within a single cross-section, can have different levels of gene expression needs to be considered and properly referenced. This has also been noted for principal cells and their expression of many different proteins (Hermo et al., 1991, 1992; Rankin et al., 1991), where in a given cross section some principal cells are highly reactive while adjacent principal cells were unreactive. In these cases, it was proposed that the principal cells were not in synchrony with respect to the secretion of these proteins. However, it may also suggest that the principal cell population of a given region is more diverse than ever imagined. Likewise, one could argue that not all the clear cells of a given region are not in synchrony with respect to the expression of a given protein at one given time or that the population of clear cells is more diverse than expected. Examples of diverse expression of GSTs in principal cells also highlights the hypothesis of diverse populations of principal cells in a given epididymal region (Papp et al., 1995). Please consider including these points in your data interpretation.

This is a very interesting question and idea, and we have added a new supplementary figure (new **Figure 3—figure supplement 1**) and Supplementary file **3** and a paragraph in the Discussion on the topic. Briefly, in principle, single cell RNA-Seq data provide some insight into cell to cell heterogeneity in gene expression, although the low efficiency of mRNA capture in scRNA-Seq makes it difficult to distinguish “true zeroes” – cases where a given cell does not express a gene – from technical dropouts resulting from failure to capture any molecules of the gene in question. To address this we used DESingle (Miao et al., 2018) to estimate the fraction of expressing cells for all genes expressed in any given principal cell subcluster, and present this analysis in new **Figure 3—figure supplement 1** and in new Supplementary file 3. Briefly, we do find that genes expressed at similar levels per cell can vary in their penetrance of expression, as shown in panels **C-E**, and consistent with the reviewer comments above. Indeed, we highlight Clusterin as one of the genes with unusually high cell to cell variability as a “positive control” – based on Hermo 1991 – in panel **B**. We anticipate that the new Supplementary file **3** should provide a valuable resource for investigators interested in the question of principal cell heterogeneity.

As an important aside, we note that cell to cell variability in apparent expression of marker genes is independent for different genes that mark a given cluster. In other words, both *Car4* and *Gdf15* are markers for Cluster 6 (new terminology) in **Figure 3D**. While these and other marker genes are not present in every cell in this Cluster, the cells lacking *Car4* and the cells lacking *Gdf15* are uncorrelated (not ANTIcorrelated, UNcorrelated). This means that there is no coherent way to subdivide Cluster 6 into meaningful subclusters – e.g. putative Clusters 6A (Car+/Gdf+) and 6B (Car-/Gdf-) – and is the reason Cluster 6 is not further subdivided. So while staining studies might find Car4+ and Car4- cells, these different populations are not indicators of cell states with larger underlying gene regulatory programs. We have attempted to efficiently express these thoughts in the revised Discussion.

11) The description of a potential new population of basal cells is interesting but no indication is given as to whether these were restricted/enriched in a particular epididymal region. Please provide this information.

We indicate the locations of all basal cells in new **Figure 5B**, revealing that the “low UMI” basal cells are found throughout the epididymis.

12) Basal cells have been suggested to be derived from the monocyte/macrophage lineage (Seiler et al., 1998, 1999). The group of Breton has indicated that basal cells are COX-1 and cytokeratin 5+ cells and distinct from the F4/80 macrophages. In addition, dendritic cells are present in the epithelium. Furthermore, all of these cells are lodged at the base of the epithelium where they firmly adhere to the basement membrane. Without the use of appropriate markers to separate these cells, how can we be certain that what is being called basal cells are not in fact a mixture of the monocyte/macrophage basal cells of the Cooper group as compared with the dendritic cells of the Breton group. LM images and quantification would appear to be a way to confirm what basal cell type was extracted.

This is an interesting point which we had neglected. We have now addressed the distinction between basal cells and macrophages/dendritic cells two ways. First, using our scRNA dataset we have grouped putative basal and immune cells, and reclustered this subset of cells. New **Figure 5—figure supplement 1A-B** shows this clustering, confirming the major transcriptional separation of basal cells and macrophage/DC cell groups. To further validate these distinctions in vivo, we also carried out two-color immunostaining to compare basal cells and macrophages. For basal cells we used the classic marker *Krt5*, and also stained with Adm to confirm the basal cell expression of this marker predicted by our scRNA-Seq dataset. For immune cells, we stained with MHC Class II for general antigen presenting cells, and F4/80 to more specifically stain macrophages – we also attempted to stain for DCs using Cd207 but the antibody proved nonspecific. Consistent with other recent studies (Shum et al., 2014), we find *Krt5*-positive basal cells generally do not stain with MHC Class II or F4/80 – these antigens stain distinct cell types – thus supporting the distinction between basal cells and mononuclear phagocytic cells. These data are show in the new **Figure 5—figure supplement 1C-D**.

13) Although in this study, macrophages are indicated to be strongly linked to the vas deferens, no indication of epididymal region is given for the other immune cell populations. Furthermore, in the heading of immune cells, the authors define a category of endothelial cells. There is no evidence of any endothelial cells being present in the epididymal epithelium. This suggests that the preparations are contaminated. Endothelial cells are found lining blood vessels and these do not exist in the epithelium. Can the authors clarify from where the endothelial cells were derived? As for the immune cells, once again some LM images and quantification of percent of each of the different immune cells examined would be in order, e.g. macrophages, dendritic cells, T-lymphocytes, etc).

The reviewer is correct to note that endothelial cells are not present in the epithelium proper – the lining of the epithelial tube. However, endothelial tubes *are* present in the stroma of the tissue, between the capsule of the vas deferens and the epithelium. We readily find capillaries in the stroma of the vas – although we would be happy to include these images as part of the rebuttal, we do not imagine this will be viewed as controversial. Anyway, we would not characterize this as “contamination”, any more than the presence of muscle cells – which did not seem to concern the reviewer – in the vas is contamination.

As for the immune cells, the location of the immune cell populations was shown in **Figure 5A** (new **Figure 6A**) – we tend to find more immune cells in the vas, but all four immune cell subclusters include representatives from throughout the epididymis. As for follow-up, as noted above we now present IF studies showing the presence of macrophages and other antigen-presenting cells, using F4/80 and MHC Class II for markers. For both of these cell types, we clearly see infiltrating cells far from any endothelial vessels, as seen in **Figure 5—figure supplement 1C-D**. This is consistent with prior staining studies which identified infiltrating macrophages in the epididymal interstitium. As far as quantitating the fractions of different immune cells in different epididymal regions, this has been done by FACS and published (Voisin et al., 2018) and we did not feel repeating this analysis by IF would much improve our single-cell RNA-Seq manuscript.

14) The cell:cell communication aspect shown by bioinformatics comparisons in Figure 6 is interesting. However, the speculation that basal cells and endothelial cells, and macrophages and "secretory fibroblasts", in the vas express complementary sets of ligands and receptors needs to be validated by another method, e.g., immunohistochemistry, to demonstrate that these cells are co-localized. Please also define PC , DC, SI, NS.

As requested, we have clarified the cell populations in this figure. Although we did a fair amount of staining of the vas to explore these cell populations, given that many of the ligand-receptor interactions involve secreted ligands that can diffuse away from the secreting cell, we do not think juxtaposition of two cells is the appropriate test of these hypotheses. We contend that future genetic studies will be needed to test the hypotheses we raise, and we state as much in the text.

15) Analysis of the vas deferens is of interest as this part of the duct is rarely studied. The authors note high expression of genes involved in mitochondrial energy metabolism. Is it possible that the cells isolated from the vas include spermatozoa? The tissue samples are from 10-14 weeks mice, when males are already sexually mature. Though the single cell isolation protocol stresses the removal of sperm from tissue, it appears to rely on visual inspection of supernatant turbidity. Some other markers should be investigated to exclude the possibility of contaminating sperm.

As requested, we have looked into expression of sperm markers (*Prm1*, etc.) in the vas deferens clusters and indeed throughout our dataset (see new **Figure 2—figure supplement 2B**), finding no evidence that the vas deferens cells in question are sperm. This is consistent with our expectation that microfluidic droplets are unlikely to capture cells as long as sperm (we tried to minimize sperm contamination largely to avoid clogging the 10X Genomics platform, not to avoid reading RNAs in sperm). In addition, the high expression of nuclear-encoded mitochondrial genes is found in cells that otherwise exhibit all other hallmarks of being epithelial cells (high level *Spink8* expression etc.), and in any case would not be expected in sperm, which have abundant mitochondria but little evidence for abundant mRNAs associated with mitochondrial biogenesis.

16) While the fluorescent images are good, they do not cover all regions of the duct, and there is no way of knowing if the images are representative. How many animals were used and how many experiments were done to ensure reproducibility?

We now have added a great deal more IF data, as detailed above in the introduction to the rebuttal. All staining efforts represent multiple slides from at least three different animals.

17) The Discussion is remarkably incomplete and somewhat biased. Some specific questions that are not satisfactorily addressed are:1) To which cell type do the so-called apical cells belong? Apical cells are not discussed at all in this manuscript;2) The discussion on immune cells is poor considering the recent publications on murine epididymal immune cells from several laboratories. Please refer to and discuss the studies focussed on epididymal immune cells, e.g., Voisin et al., 2018;3) There is a complete lack of discussion of the major and well-known roles of epididymal sperm maturation concerning the fine balance between oxidative and anti-oxidative processes. These are just a few examples, as there are many landmark manuscripts dealing with epididymal physiology that have been ignored.

We have attempted to address all of these comments in the revised manuscript (often in the relevant Results section rather than Discussion, to avoid duplicating text). That said, in our opinion it is somewhat unfair to call the Discussion “remarkably incomplete” – the point of Discussion sections is generally to cover some of the implications arising from key points of a manuscript, not to review the entire literature on the subject. Especially for a genomics paper, it is simply not feasible to list every paper that studied expression of a gene by q-RT-PCR or IF in this tissue, as one might infer to be the intention of the reviewer’s comment here.

More broadly, we do not view it as an obligation to cite and discuss every “landmark manuscript dealing with epididymal physiology” – this would require writing a textbook, not a manuscript. The purpose of this Resource paper is not to provide a comprehensive list of every paper ever published on this tissue. When we cite prior publications, the purpose is to provide sufficient proof of concept, and validation by comparison of our new data to known facts, to provide confidence in our dataset. Beyond that, our goal is to discuss features that we find interesting, with some balance between summarizing prior art and suggesting directions for future efforts.

Nonetheless, we have addressed the following items in the revised manuscript (either in the Discussion or the Results section) to better fulfill the reviewers’ wishes:

1) We have further elaborated on the relationship between our clear cell data, and particularly the Amylin-positive clear cell subset, and both narrow and apical cells. Based on the localization of our Amylin-positive clear cells (extending into the proximal corpus) as well as the presence of Amylin-negative cells with narrow cell morphology in the initial segment, we do not favor the hypothesis that Amylin is a marker for narrow or apical cells. As the other two clear cell clusters include representatives from throughout the epididymis, we therefore conclude that our dataset does not include molecular evidence of specific markers for narrow or apical cells.

2) We have modestly expanded on the balance between oxidation and reduction activities, as requested

3) We have added a paragraph noting the similarities between cellular organization in the kidney and in the epididymis, as requested in other comments

4) We have added a half paragraph describing the new analysis of cell to cell variation, as requested in other comments

5) We have explored the question of the relationship between basal cells and mononuclear phagocytes with new data, and we have attempted to better contextualize our findings regarding immune populations in the epididymis in the Results section.

In addition, the speculation in the Discussion on the highly segmental expression of several large multigene families and its relative importance is "epididymocentric". It could be argued that this is a feature of any tubular structure in the body and the authors are encouraged to examine (for example) the single cell data from the mouse kidney before making the assumption that this is an epididymis-specific feature.

This is a very interesting point, as many features of the epididymis are indeed also observed in kidney tubular epithelium. As requested, we now add a Discussion paragraph comparing epididymal gene expression patterns along the tubule to gene regulation in the kidney.